# Plasticity-induced actin polymerization in the dendritic shaft regulates intracellular AMPA receptor trafficking

**Victor C Wong\*, Patrick R Houlihan, Hui Liu, Deepika Walpita, Michael C DeSantis, Zhe Liu, Erin K O'Shea**

Janelia Research Campus, Howard Hughes Medical Institute, Ashburn, United States

**Abstract** AMPA-type receptors (AMPARs) are rapidly inserted into synapses undergoing plasticity to increase synaptic transmission, but it is not fully understood if and how AMPAR-containing vesicles are selectively trafficked to these synapses. Here, we developed a strategy to label AMPAR GluA1 subunits expressed from their endogenous loci in cultured rat hippocampal neurons and characterized the motion of GluA1-containing vesicles using single-particle tracking and mathematical modeling. We find that GluA1-containing vesicles are confined and concentrated near sites of stimulation-induced structural plasticity. We show that confinement is mediated by actin polymerization, which hinders the active transport of GluA1-containing vesicles along the length of the dendritic shaft by modulating the rheological properties of the cytoplasm. Actin polymerization also facilitates myosin-mediated transport of GluA1-containing vesicles to exocytic sites. We conclude that neurons utilize F-actin to increase vesicular GluA1 reservoirs and promote exocytosis proximal to the sites of synaptic activity.

## Editor's evaluation

In this manuscript, the authors developed a sensitive single particle tracking method for endogenous AMPA receptors. They found that AMPAR-containing vesicles showed reduced mobility near stimulation sites, due to increased F-actin bundling in dendritic shafts. The study provides compelling evidence on a new important mechanism of AMPAR trafficking using state-of-the-art labeling and analysis techniques.

\*For correspondence:
wongv@janelia.hhmi.org

## Introduction

Synaptic plasticity – the modulation of synaptic connections in response to changes in neuronal activity – is regarded as the cellular basis for learning and memory (*Abbott and Nelson, 2000*; *Citri and Malenka, 2008*; *Magee and Grienberger, 2020*). Changes in postsynaptic spine morphology (i.e. structural plasticity) and protein composition are observed during synaptic plasticity and are thought to contribute to changes in synaptic transmission (i.e. functional plasticity; *Kasai et al., 2010*; *Nakahata and Yasuda, 2018*; *Kasai, 2023*). Persistent pre-synaptic neurotransmitter release stimulates postsynaptic spine enlargement and insertion of proteins into the synapse (*Matsuzaki et al., 2004*; *Okamoto et al., 2004*; *Hayashi and Majewska, 2005*; *Patterson and Yasuda, 2011*), including α-amino-3-hydroxy-5-methyl-4-isoxazolepropionic acid receptors (AMPARs; *Song and Huganir, 2002*) – ionotropic glutamate receptors that permit the influx of sodium, potassium, and in certain cases calcium ions (*Chater and Goda, 2014*). This increase in AMPAR abundance, as well as changes to AMPAR conductance, results in a greater inward flow of ions in response to stimulation (*Anggono and Huganir, 2012*; *Diering and Huganir, 2018*).

Given that the number of AMPARs at synapses is highly regulated during synaptic plasticity, AMPAR trafficking has been studied extensively (*Malinow and Malenka, 2002*; *Henley et al., 2011*; *Park, 2018*; *Diering and Huganir, 2018*; *Groc and Choquet, 2020*; *Díaz-Alonso and Nicoll, 2021*). The majority of AMPARs are synthesized in the cell body and then enter the secretory pathway, where they are trafficked in vesicles and inserted into the neuronal membrane via exocytosis (*Shepherd and Huganir, 2007*). AMPARs on the membrane can diffuse into synapses (*Choquet and Opazo, 2022*) and can also be endocytosed and delivered to lysosomes for degradation or recycled through the endocytic pathway (*Ehlers, 2000*; *Carroll et al., 2001*; *Goo et al., 2017*; *Hanley, 2018*). An important feature of synaptic plasticity is input specificity – plasticity stimulated at one synapse does not spread to inactive synapses (*Kandel et al., 2013*) – suggesting that AMPARs are inserted selectively into stimulated synapses. The mechanisms that regulate how AMPARs are delivered to synapses undergoing plasticity are not fully understood, but evidence points to two models: (1) AMPARs in vesicles are trafficked and exocytosed directly into or adjacent to stimulated synapses (*Gerges et al., 2006*; *Kennedy et al., 2010*; *Cho et al., 2015*; *Park, 2018*); and (2) AMPARs diffusing in the neuronal membrane are selectively trapped at stimulated synapses (*Borgdorff and Choquet, 2002*; *Tardin et al., 2003*; *Petrini et al., 2009*; *Ehlers et al., 2007*; *Opazo et al., 2010*; *Opazo and Choquet, 2011*; *Penn et al., 2017*; *Choquet and Opazo, 2022*).

AMPARs in vesicles are trafficked through dendrites by both microtubule- and actin-based motors, and disrupting active transport of AMPARs interferes with long-term potentiation (LTP), a commonly studied form of plasticity (*Setou et al., 2002*; *Hoogenraad et al., 2005*; *Correia et al., 2008*; *Wang et al., 2008*; *Hoerndli et al., 2013*; *Hoerndli et al., 2015*; *Wagner et al., 2019*; *Gutiérrez et al., 2021*). Nevertheless, it is unclear whether AMPAR-containing vesicles (hereafter referred to as AMPAR vesicles) are delivered directly to synapses undergoing plasticity, due in large part to limitations in imaging AMPARs in vesicles as opposed to on cell membranes (*Groc and Choquet, 2020*). One approach to studying AMPAR vesicles utilizes chemically inducible dimerization to control the release of exogenous AMPARs (i.e. AMPARs expressed from plasmid DNA) from the endoplasmic reticulum into the secretory pathway, followed by tracking AMPAR vesicles as they traverse photobleached sections of dendrite (*Hangen et al., 2018*; *Bonnet et al., 2023*). Using this technique, *Hangen et al., 2018* found AMPAR vesicles slow down and pause in response to elevated intracellular calcium levels during neuronal activity, and consequently hypothesized that a calcium-mediated mechanism primes AMPAR vesicles for exocytosis. However, it is unclear if pausing is directly linked to AMPAR exocytosis at synapses undergoing plasticity. Additional studies have demonstrated that endosomes containing exogenous AMPARs can enter dendritic spines (*EstevesdaSilva et al., 2015*; *Bowen et al., 2017*), raising the possibility of direct AMPAR exocytosis into synapses. However, scanning electron micrographs of immunogold-labeled AMPARs fail to reveal a substantial fraction of AMPAR vesicles in spines (*Tao-Cheng et al., 2011*). Moreover, imaging exogenous AMPARs tagged with super ecliptic pHluorin shows that exocytosis occurs largely at extrasynaptic sites (*Lin et al., 2009*; *Makino and Malinow, 2009*; *Patterson et al., 2010*), favoring a model in which receptors diffuse into synapses after exocytosis.

Much research has been focused on understanding how synapses capture AMPARs as they diffuse through the neuronal membrane (*Opazo and Choquet, 2011*; *Groc and Choquet, 2020*; *Choquet and Opazo, 2022*). After exocytosis, AMPARs diffuse freely in random directions, but diffusion decreases precipitously at synapses (*Tardin et al., 2003*) because AMPARs are anchored there by postsynaptic density (PSD) proteins, such as PSD-95 (*El-Husseini et al., 2000*; *Bats et al., 2007*; *Opazo et al., 2010*; *Opazo et al., 2012*; *Chen et al., 2015*). Synaptic activity changes both the composition of proteins in the synapse and posttranslational modifications on AMPARs, further enhancing receptor anchoring (*Opazo et al., 2010*; *Diering and Huganir, 2018*; *Lu and Roche, 2012*; *Opazo et al., 2012*). These observations support a model where AMPARs may not be trafficked to specific loci, but rather diffuse in random directions, only to be concentrated in active synapses as a consequence of their increased residence time. Importantly, crosslinking AMPARs on the neuronal membrane to prevent their diffusion impairs synaptic potentiation in vivo (*Penn et al., 2017*; *Getz et al., 2022*). Nevertheless, the net distance a receptor can travel via diffusion is limited (*Groc and Choquet, 2020*). Consequently, this model depends on the presence of nearby extrasynaptic reservoirs from which synapses can draw AMPARs during plasticity (*Choquet and Opazo, 2022*). How reservoirs are established and maintained is not fully understood, but given that synapses can be located hundreds of

microns from the cell body, it is probable that receptors are actively transported near sites undergoing synaptic plasticity.

To address whether and how neurons specify the location to which AMPAR vesicles are delivered, we developed a method to identify vesicles containing AMPAR GluA1 subunits expressed at native levels from endogenous loci and characterize the motion of these vesicles in cultured rat hippocampal neurons. Using this technique, we identify previously undescribed motion behaviors for GluA1-containing vesicles (hereafter referred to as GluA1 vesicles). We show that stimulating synaptic activity with glycine-induced chemical LTP (cLTP) or structural plasticity with glutamate uncaging-evoked structural LTP (sLTP) results in the local confinement of GluA1 vesicles in the dendritic shaft. We find that confinement concentrates GluA1 vesicles near sites of stimulation, thereby increasing the size of GluA1 reservoirs near these sites. GluA1 vesicle confinement is the result of stimulation-induced actin polymerization in the dendritic shaft, which changes the rheological properties of the dendritic cytoplasm in a manner that inhibits transport along the length of the dendrite and inhibits diffusion of GluA1 vesicles. Finally, we show that actin polymerization in the dendritic shaft near the sites of stimulation facilitates myosin-mediated transport of GluA1 vesicles from intracellular reservoirs to sites of exocytosis. In sum, our results suggest that neurons enhance the delivery AMPARs to active synapses by restricting the motion of AMPAR vesicles away from these synapses while simultaneously promoting AMPAR exocytosis near these synapses.

## Results

### GluA1-HaloTag is trafficked to postsynaptic densities and responds to stimulation

To study the intracellular transport of AMPARs during neuronal activity, we developed a method to label endogenous AMPAR GluA1 subunits (encoded by *Gria1*), expressed at native levels, using homology-independent targeted integration (HITI; *Suzuki et al., 2016*) in cultured rat hippocampal neurons. HaloTag (HT; *Los et al., 2008*) enables labeling of GluA1 with bright and photostable Janelia Fluor (JF) dyes (*Grimm et al., 2015*), which can be conjugated to the HaloTag ligand (HTL). We used HITI to insert HaloTag into the extracellular amino-terminal domain (NTD) of GluA1 at R280 (*Figure 1A* and *Figure 1—figure supplement 1A*), resulting in a high knock-in efficiency (*Figure 1—figure supplement 1B–C*). GluA1 edited at this position with HaloTag (GluA1-HT) and labeled with JF$_{549}$-HaloTag ligand (JF$_{549}$-HTL) is concentrated in dendritic spines, similar to endogenous GluA1 (*Figure 1B*; *Craig et al., 1993*). We find that the majority of edited sequences contain HaloTag in the correct orientation and without indels (*Figure 1—figure supplement 1D*), and that HaloTag insertion does not alter the expression of *Gria1* (*Figure 1—figure supplement 2A–B*). We observe correlated localization of HaloTag and *Gria1* mRNA in the same neuron (*Figure 1—figure supplement 2C*), indicating that *Gria1*-HaloTag is produced as a single transcript. Using stimulation emission depletion (STED) microscopy to examine GluA1 and HaloTag labeling at subdiffraction-limited length scales, we find that HaloTag signal has a strong overlap with native GluA1 signal (*Figure 1—figure supplement 3A*), indicating that GluA1-HT is translated as a single peptide sequence. Using immunofluorescence labeling and STED, we demonstrate that HaloTag insertion at R280 does not disrupt GluA1 trafficking to postsynaptic densities (*Figure 1—figure supplement 3B–D*). Finally, using electrophysiological methods, we demonstrate that insertion of HaloTag at R280 in the NTD of GluA1 does not significantly alter its channel function (*Figure 1—figure supplement 4*).

### Neuronal activity confines GluA1 vesicles by disrupting vesicle motion in the dendritic shaft of cultured rat hippocampal neurons

To overcome the challenge of signal saturation (*Figure 1B*) and track single GluA1-HT vesicles inside the dendritic shaft, we developed a block-and-chase protocol to achieve sparse labeling of de novo synthesized endogenous GluA1 (*Figure 1C*). To avoid visualizing pre-existing GluA1-HT, GluA1-HT was first labeled with a saturating concentration of JF$_{646}$-HTL in the presence of the translation inhibitor cycloheximide (CHX). Next, JF$_{646}$-HTL and CHX were washed away, and after an incubation period to allow GluA1-HT translation to recover, de novo synthesized GluA1-HT was labeled with JF$_{549}$-HTL. Because JF$_{549}$-HTL labels only newly synthesized GluA1-HT, signal detected by fluorescence microscopy is dramatically reduced in this fluorescence channel, allowing us to identify sparse, punctate

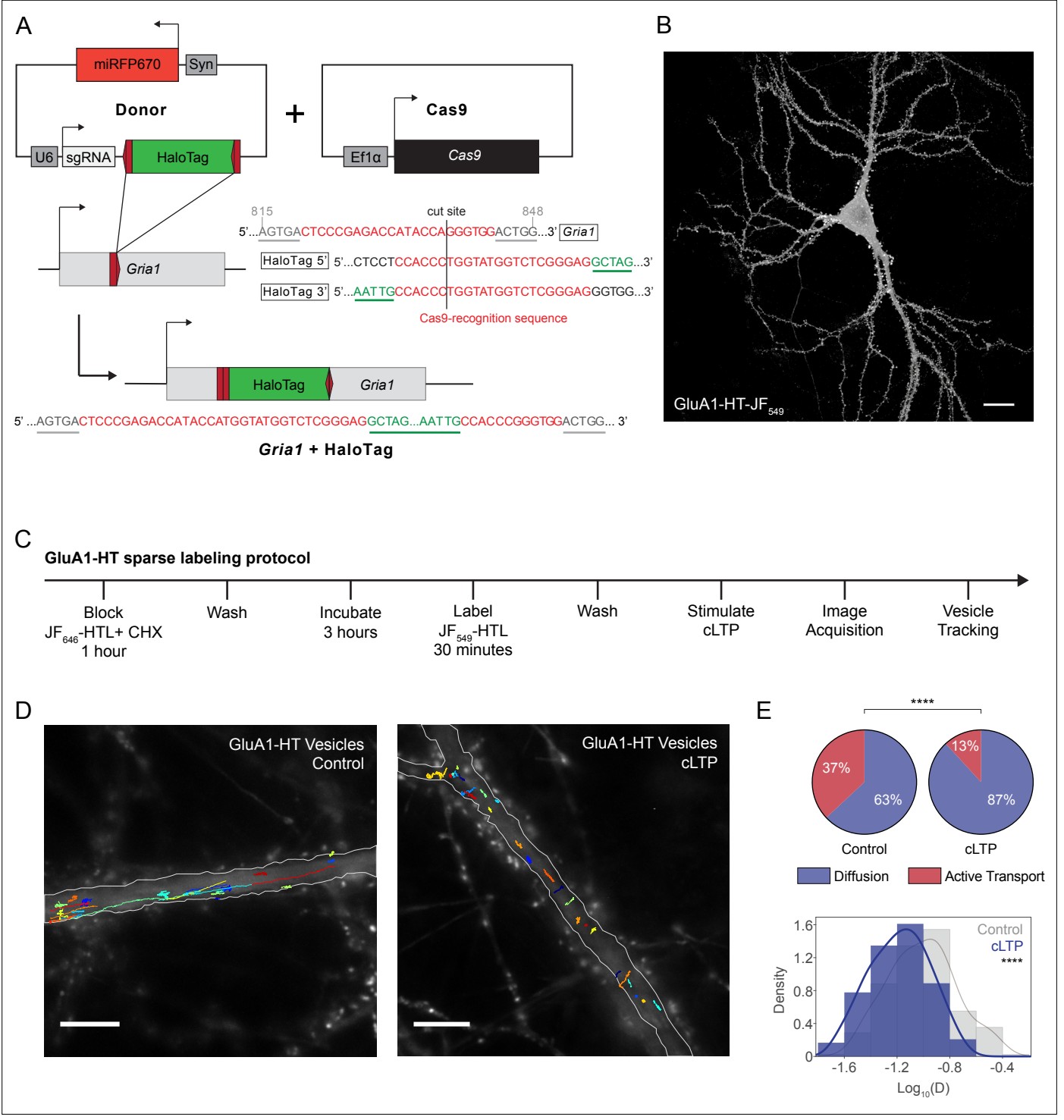

**Figure 1.** Chemical LTP induction reduces active transport and diffusion of GluA1-HT vesicles in the dendritic shaft. (**A**) Schematic of *Gria1* gene targeting with HaloTag (HT) by homology-independent targeted integration (HITI). A donor construct for HaloTag integration (Donor) is co-transfected into rat hippocampal neurons with a construct expressing Cas9 (Cas9). The donor contains HaloTag flanked on each side by one copy of the *Gria1* sequence to be targeted by Cas9, a single guide RNA (sgRNA), and a neuronal transfection marker (e.g. miRFP670). In neurons transfected with both the Cas9 and Donor constructs, Cas9 creates a double strand break in the genomic copy of *Gria1* and also excises the HaloTag sequence from the donor construct. The freed HaloTag sequence can be inserted into the genomic cut site when the double strand break is repaired by non-homologous end joining (NHEJ). (**B**) Representative confocal image of a cultured rat hippocampal neuron expressing endogenous GluA1 tagged with HaloTag and labeled with JF$_{549}$-HaloTag ligand (GluA1-HT-JF$_{549}$). Scale bar, 20 µm. (**C**) Experimental workflow to achieve sparse GluA1-HT labeling for GluA1-HT vesicle identification and single-particle tracking (SPT) analysis. (**D**) GluA1-HT vesicle trajectories in a dendritic shaft with no treatment (Control)

*Figure 1 continued on next page*

*Figure 1 continued*

and during cLTP induction (cLTP). Trajectories are overlaid on epifluorescence images of GFP-Homer1c, which was used to identify and segment the dendritic shaft. Scale bar, 10 µm. For videos, see *Figure 1—video 5* and *Figure 1—video 6*. (**E**) Pie charts: fractions of vesicles exhibiting diffusion or active transport in dendritic shafts with no treatment (Control) and during cLTP induction (cLTP). ****p<0.0001 by Mann-Whitney test. n=12–14 timelapse imaging sequences (each timelapse captures the motion of GluA1-HT vesicles in one region of dendrite in one neuronal culture) for each condition. Histogram: distributions of diffusion coefficients for GluA1-HT vesicles without treatment (Control; gray) and during cLTP induction (cLTP; blue). Line represents the probability density function of each histogram estimated by kernel density estimation (KDE). ****p<0.0001 by Kolmogorov-Smirnov test. n=227–360 GluA1-HT vesicle trajectories pooled from 12 to 14 timelapses for each condition.

The online version of this article includes the following video, source data, and figure supplement(s) for figure 1:

**Source data 1.** Related to *Figure 1*.

**Figure supplement 1.** Successful insertion of HaloTag into *Gria1* depends on target site and linker length.

**Figure supplement 1—source data 1.** Related to *Figure 1—figure supplement 1*.

**Figure supplement 2.** Identification and quantification of *Gria1*-HaloTag mRNA transcripts using HCR RNA-FISH.

**Figure supplement 2—source data 1.** Related to *Figure 1—figure supplement 2*.

**Figure supplement 3.** HaloTag is correctly inserted into the N-terminal domain of GluA1 and GluA1-HT is trafficked to postsynaptic densities.

**Figure supplement 3—source data 1.** Related to *Figure 1—figure supplement 3*.

**Figure supplement 4.** Currents elicited by GluA1-HT are similar to currents elicited by untagged GluA1.

**Figure supplement 4—source data 1.** Related to *Figure 1—figure supplement 4*.

**Figure supplement 5.** HMM-Bayes analysis can be used to infer the motion states of GluA1-HT-JF$_{549}$ particles and determine motion parameters.

**Figure supplement 5—source data 1.** Related to *Figure 1—figure supplement 5*.

**Figure supplement 6.** Identifying GluA1-HT vesicles based on their bleaching characteristics and motion.

**Figure supplement 6—source data 1.** Related to *Figure 1—figure supplement 6*.

**Figure supplement 7.** GluA1-HT-JF$_{549}$ receptors trapped in postsynaptic densities have very low diffusion coefficients.

**Figure supplement 7—source data 1.** Related to *Figure 1—figure supplement 7*.

**Figure supplement 8.** Block-and-chase protocol enables sparse labeling of GluA1-HT and the identification of GluA1-HT vesicles.

**Figure supplement 9.** cLTP stimulation results in increased surface labeling of GluA1-HT by JF$_{549}$i-HTL.

**Figure supplement 9—source data 1.** Related to *Figure 1—figure supplement 9*.

**Figure supplement 10.** GluA1-HT vesicles exhibit subdiffusive motion during cLTP induction.

**Figure supplement 10—source data 1.** Related to *Figure 1—figure supplement 10*.

**Figure supplement 11.** cLTP induction increases the probability GluA1-HT vesicles switch from active transport to diffusion.

**Figure supplement 11—source data 1.** Related to *Figure 1—figure supplement 11*.

**Figure 1—video 1.** Timelapse sequence of GluA1-HT-JF$_{549}$ after block-and-chase labeling with JF$_{646}$-HTL and JF$_{549}$-HTL in the dendrite of a cultured rat hippocampal neuron.
https://elifesciences.org/articles/80622/figures#fig1video1

**Figure 1—video 2.** Timelapse sequence of GluA1-HT-JF$_{549}$ after block-and-chase labeling with JF$_{646}$-HTL and JF$_{549}$-HTL in a dendrite that has been treated with Nocodazole.
https://elifesciences.org/articles/80622/figures#fig1video2

**Figure 1—video 3.** Timelapse sequence of GluA1-HT-JF$_{549}$i on the surface of a dendrite after block-and-chase labeling with JF$_{646}$-HTL and JF$_{549}$i-HTL.
https://elifesciences.org/articles/80622/figures#fig1video3

**Figure 1—video 4.** Timelapse sequence of GluA1-HT-JF$_{549}$ after block-and-chase labeling with JF$_{646}$-HTL and JF$_{549}$-HTL, with GluA1-HT-JF$_{549}$ particles that colocalize with GFP-Homer1c highlighted.
https://elifesciences.org/articles/80622/figures#fig1video4

**Figure 1—video 5.** Timelapse sequence of GluA1-HT-JF$_{549}$ after block-and-chase labeling with JF$_{646}$-HTL and JF$_{549}$-HTL in a dendrite with no stimulation.
https://elifesciences.org/articles/80622/figures#fig1video5

**Figure 1—video 6.** Timelapse sequence of GluA1-HT-JF$_{549}$ after block-and-chase labeling with JF$_{646}$-HTL and JF$_{549}$-HTL in a dendrite during cLTP induction.
https://elifesciences.org/articles/80622/figures#fig1video6

GluA1-HT-JF$_{549}$ conjugates (*Figure 1—video 1*). We then used single-particle tracking (SPT) analysis to reconstruct trajectories, and hidden Markov modeling with Bayesian model selection (HMM-Bayes; *Monnier et al., 2015*) to determine important motion parameters (i.e. velocity and diffusion coefficient) and predict the motion type (i.e. active transport versus diffusion) of each trajectory (*Figure 1—figure supplement 5*; *Jaqaman et al., 2008*; *Liu et al., 2018*). Finally, we separated GluA1-HT vesicles from surface GluA1-HT based on motion type, fluorescence bleaching characteristics, and localization in the dendritic shaft (*Figure 1—figure supplements 6–8*).

Having established a pipeline to label, track, and analyze GluA1-HT vesicles, we sought to evaluate whether the motion of GluA1-HT vesicles is modulated by synaptic activity. Glycine-induced chemical LTP (cLTP) is an established method of stimulating synaptic plasticity and increasing the surface expression of GluA1 in cultured neurons (*Lu et al., 2001*; *Passafaro et al., 2001*; *Park et al., 2006*; *Molnár, 2011*). We hypothesized that cLTP induction might change the motion of GluA1-HT vesicles in the dendritic shaft to support increased GluA1-HT exocytosis during neuronal activity. Using a membrane impermeable variant of JF$_{549}$-HTL termed JF$_{549}$i-HaloTag ligand (JF$_{549}$i-HTL; 'i' for impermeant; *Xie et al., 2017*), we validated that cLTP stimulates increased expression of GluA1-HT on neuronal surfaces (*Figure 1—figure supplement 9*). We then imaged and tracked GluA1-HT vesicles in dendrites during cLTP induction, and observed clear qualitative differences in motion compared to vesicles in the dendrites of unstimulated control neurons (*Figure 1D*, *Figure 1—video 5* and *Figure 1—video 6*). Most strikingly, we observed a loss in long-range motion along the length of the dendritic shaft in cLTP-stimulated neurons. When HMM-Bayes is applied to characterize trajectories collected during cLTP induction, we find a significant decrease in the fraction of GluA1-HT vesicles undergoing active transport (*Figure 1E*, pie charts). The diffusion coefficients (D) for GluA1-HT vesicles undergoing diffusion are also significantly reduced by cLTP induction (*Figure 1E*, histogram). In addition, vesicles exhibit increased subdiffusion (i.e. constrained diffusion due to molecular crowding or interactions; *Feder et al., 1996*; *Saxton, 2007*) in response to cLTP induction (*Figure 1—figure supplement 10*).

These observations demonstrate that the overall motion of GluA1-HT vesicles is inhibited by cLTP, suggesting that vesicle motion is locally confined (defined in this work as the restriction of vesicle motion away from its initial position). We reasoned that if cLTP spatially confines GluA1-HT vesicles then it should, in addition to decreasing the fraction of vesicles undergoing active transport, prevent diffusing vesicles from transitioning to active transport and leaving their local regions. A critical feature of HMM-Bayes is the ability to infer multiple motion states from a single trajectory and determine state-transition probabilities (*Figure 1—figure supplement 11A–B*; *Monnier et al., 2015*). Using HMM-Bayes, we find multi-state GluA1-HT vesicles stochastically switch motion states from active transport to diffusion and from diffusion to active transport with approximately the same probability under unstimulated control conditions (*Figure 1—figure supplement 11C*, Control, $k_{AT-D}$ vs $k_{D-AT}$). By contrast, during cLTP, GluA1-HT vesicles have a greater probability of switching from active transport to diffusion than from diffusion to active transport (*Figure 1—figure supplement 11C*, cLTP, $k_{AT-D}$ vs $k_{D-AT}$). Furthermore, GluA1-HT vesicles undergoing diffusion have a high probability to continue diffusing (*Figure 1—figure supplement 11C*, cLTP, $k_{D-D}$). Taken together, our observations suggest that cLTP induction results in the local confinement of GluA1-HT vesicles.

## Local induction of synaptic activity confines GluA1 vesicles near the site of activity by disrupting GluA1 vesicle motion

If confinement is a mechanism to increase the intracellular reservoir of GluA1-HT near sites of synaptic activity, then stimulating plasticity at a specific synapse should alter GluA1-HT vesicle motion near that synapse. Single-photon (1 P) 4-Methoxy-7-nitroindolinyl-caged-L-glutamate (MNI) uncaging can be used to stimulate synaptic activity in a desired region (*Ellis-Davies, 2007*). We used a glutamate-uncaging evoked structural LTP protocol (sLTP; *Matsuzaki et al., 2004*) to induce structural plasticity at a spine of interest. This protocol has been shown to stimulate N-methyl-D-aspartate receptor (NMDAR)-mediated calcium transients, which result in spine expansion and increased AMPAR concentration at the targeted synapse – two important proxies for functional plasticity (*Matsuzaki et al., 2001*; *Matsuzaki et al., 2004*; *Lee et al., 2009*; *Patterson and Yasuda, 2011*; *Huganir and Nicoll, 2013*; *Bosch et al., 2014*; *Kruijssen and Wierenga, 2019*). First, we calibrated the strength of the laser so that it would not trigger calcium influx or structural plasticity in neighboring spines. In targeted

spines, we observed a significant increase in the area of the spine head and in GluA1-HT after stimulation in dishes containing MNI, but not in dishes without MNI (*Figure 2A* and *Figure 2—figure supplements 1–2*). The increase in spine size persists 10 min after the cessation of sLTP, indicating that this protocol induces sustained changes to activity (*Figure 2—figure supplement 3A*, images and Spine area line graph).

To examine GluA1-HT vesicle motion proximal to the site of structural plasticity, we separated the dendritic shaft longitudinally (i.e. along the length of the dendrite) into three equal zones based on proximity to the site of uncaging and assessed the different types of motion that GluA1-HT vesicles exhibit in each zone after sLTP (*Figure 2B* and *Figure 2—video 1*). We find that sLTP results in reduced active transport and lower rates of diffusion for GluA1-HT vesicles in Zone 1 but not Zone 2 or Zone 3 (*Figure 2C–D*), indicating that stimulation disrupts GluA1-HT vesicle motion proximal, but not distal, to the site of synaptic activity. By contrast, sLTP stimulation in the absence of MNI failed to alter the fraction of vesicles exhibiting active transport or the diffusion coefficient of vesicles both proximal and distal to the site of stimulation (*Figure 2—figure supplement 3B–C*), demonstrating that changes in vesicle movement after sLTP are not the result of laser exposure. Furthermore, GluA1-HT vesicles in Zone 1 imaged 10 min after the cessation of sLTP stimulation exhibited similar motion behaviors to those imaged immediately after the cessation of sLTP (*Figure 2—figure supplement 3A*, Active transport bar graph and Diffusion histogram), suggesting that vesicles are confined.

To further test whether vesicles are confined near the site of synaptic activity, we used HMM-Bayes to determine the probabilities that vesicles switch between active transport and diffusion in each zone during sLTP. sLTP increases the probability that multi-state GluA1-HT vesicles undergoing active transport in Zone 1 switch to and stay in a diffusive state (*Figure 2—figure supplement 4*, sLTP Zone 1, $k_{D-AT}$ vs $k_{AT-D}$, and $k_{D-D}$) in a manner dependent on the presence of MNI (*Figure 2—figure supplement 4*, No MNI, $k_{D-AT}$ vs $k_{AT-D}$, and $k_{D-D}$). By contrast, multi-state vesicles in Zone 2 and Zone 3 have similar probabilities of switching between active transport and diffusion after sLTP (*Figure 2—figure supplement 4*, sLTP Zone 2+3, $k_{D-AT}$ vs $k_{AT-D}$). These observations demonstrate that multi-state GluA1-HT vesicles proximal to the site of stimulation have low probabilities of being transported away from the site of stimulation. Combined, these findings demonstrate that sLTP results in the confinement of GluA1-HT vesicles near the site of structural plasticity.

## Confinement of GluA1 vesicles during synaptic activity is mediated by F-actin-induced molecular crowding in the dendritic shaft

Having demonstrated that sLTP results in the confinement of GluA1-HT vesicles in the dendritic shaft near the site of structural plasticity, we sought to determine the molecular mechanisms involved in disrupting vesicle motion. We hypothesized that F-actin in the dendritic shaft might be involved because stimulating neuronal activity leads to elevated intracellular calcium levels and the activation of calcium signaling pathways that trigger the polymerization of actin (*Okamoto et al., 2004*; *Okamoto et al., 2007*; *Okamoto et al., 2009*). Moreover, recent studies have reported the rearrangement of F-actin networks in the dendritic shaft during neuronal activity (*Schätzle et al., 2018*; *Lavoie-Cardinal et al., 2020*), and found that F-actin networks can reposition lysosomes in neurites (*Katrukha et al., 2017*; *van Bommel et al., 2019*). Importantly, F-actin networks formed in response to sLTP are persistent (*Okamoto et al., 2004*), and therefore could be a mechanism to confine GluA1-HT vesicles even after the cessation of stimulation.

To determine whether actin polymerization occurs in the dendritic shaft of cultured rat hippocampal neurons during synaptic activity, we tested whether cLTP would lead to the redistribution of F-tractin (tractin), an actin binding peptide that is used as a marker for F-actin (*Schell et al., 2001*). Prior to cLTP, tractin is diffusely distributed in the dendritic shaft and concentrated in spines (*Figure 3A*, Control). During cLTP induction, tractin in the dendritic shaft redistributes into a network of filaments (*Figure 3A*, cLTP). The combined length of these tractin filaments in the dendritic shaft is significantly greater during cLTP (*Figure 3A*, Tractin length, and *Figure 3—figure supplement 1*), suggesting that cLTP stimulates actin polymerization in the dendritic shaft. To eliminate the possibility that changes in the distribution of tractin are due to morphological changes in the dendrite, neurons were also transduced with a plasmid expressing tdTomato, which did not dramatically redistribute during cLTP (*Figure 3—figure supplement 2A*).

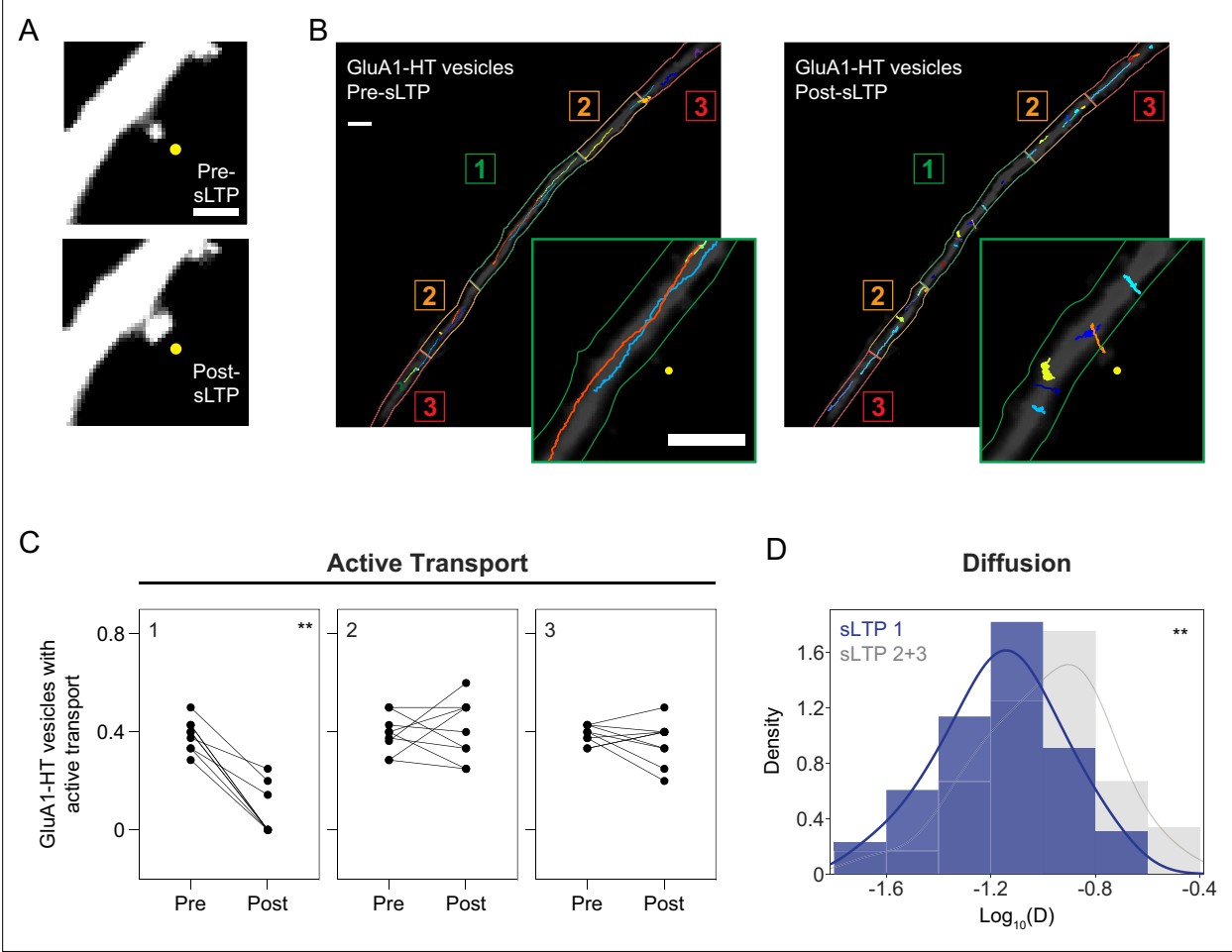

**Figure 2.** Structural LTP stimulation reduces active transport and diffusion of GluA1-HT vesicles proximal to the site of synaptic activity. (**A**) Representative epifluorescence images of a dendritic spine expressing GFP immediately before sLTP (Pre-sLTP) and after sLTP (Post-sLTP). Yellow dot indicates the site of uncaging. Scale bar, 2 μm. (**B**) GluA1-HT vesicle trajectories immediately before sLTP (Pre-sLTP) and after sLTP (Post-sLTP) overlaid on a dendrite expressing GFP. Dendrite is separated into three equal zones based on proximity to the stimulated spine to distinguish proximal and distal areas (Materials and methods). Insets: magnified images of trajectories near the site of uncaging (yellow dot). Scale bars, 5 μm. For video, see *Figure 2—video 1*. (**C**) Fractions of GluA1-HT vesicles exhibiting active transport pre- and post-sLTP in each zone. \*\*p=0.0039 by Wilcoxon matched pairs test. Each dot represents the fraction of vesicles exhibiting active transport in the indicated zone for one sLTP stimulation experiment. n=9 sLTP stimulation experiments (each experiment targets one spine on one neuron) where GluA1-HT vesicle motion in the dendrite is captured immediately before and after sLTP stimulation. (**D**) Distributions of diffusion coefficients for GluA1-HT vesicles in Zone 1 (sLTP 1; blue) versus Zone 2+3 (sLTP 2+3; gray) after sLTP stimulation. Line represents the probability density function of each histogram estimated by kernel density estimation (KDE). \*\*p=0.0037 by Kolmogorov-Smirnov test. n=60–66 GluA1-HT vesicle trajectories pooled together from nine sLTP stimulation experiments.

The online version of this article includes the following video, source data, and figure supplement(s) for figure 2:

**Source data 1.** Related to *Figure 2*.

**Figure supplement 1.** MNI uncaging calibration.

**Figure supplement 1—source data 1.** Related to *Figure 2—figure supplement 1*.

**Figure supplement 2.** sLTP stimulation increases GluA1-HT-JF$_{646}$ in targeted spines.

**Figure supplement 2—source data 1.** Related to *Figure 2—figure supplement 2*.

**Figure supplement 3.** Motion of GluA1-HT vesicles 10 min after the cessation of sLTP stimulation is similar to the motion of vesicles immediately after sLTP stimulation.

**Figure supplement 3—source data 1.** Related to *Figure 2—figure supplement 3*.

**Figure supplement 4.** sLTP increases the probability GluA1-HT vesicles switch from active transport to diffusion proximal to the site of stimulation.

**Figure supplement 4—source data 1.** Related to *Figure 2—figure supplement 4*.

**Figure 2—video 1.** Timelapse sequences of GluA1-HT-JF$_{549}$ after block-and-chase labeling with JF$_{646}$-HTL and JF$_{549}$-HTL pre-sLTP (Figure 2—video 1,

*Figure 2 continued on next page*

To confirm that the change in tractin distribution during cLTP induction is the result of actin polymerization, we imaged tractin during cLTP in dendrites that were also treated with Latrunculin A (LatA), an inhibitor of actin polymerization (*Figure 3B* and *Figure 3—figure supplement 2B*). Treating neurons with LatA not only prevents redistribution of tractin during cLTP, but also reduces the intensity of tractin signal in dendritic spines (*Figure 3B*, cLTP +LatA), demonstrating that actin polymerization leads to the redistribution of tractin during cLTP. Because the expression of actin-binding peptides may result in artificial F-actin structures (*Melak et al., 2017*), we also labeled F-actin with phalloidin after cLTP and imaged using STED microscopy. cLTP resulted in greater phalloidin labeling in dendritic shafts, while LatA treatment during cLTP decreased phalloidin labeling (*Figure 3—figure supplement 2C*), recapitulating our finding that cLTP induces actin polymerization in the dendritic shaft.

We next sought to determine whether sLTP-mediated changes in local actin networks play a role in positioning vesicles near sites of stimulation. We observed a significant increase in tractin fluorescence (MFI) in spines stimulated with sLTP and in the dendritic shaft proximal to these spines, reflecting increased actin polymerization at these locations (*Figure 3C*, blue outline). Tractin signal increases in an approximately 30 µm longitudinal section along the length of the dendritic shaft surrounding the sLTP-stimulated spine (*Figure 3—figure supplement 3A*). The increase in tractin MFI during sLTP is dependent on actin polymerization and is not an artifact of photostimulation (*Figure 3—figure supplement 3B*).

Having found that sLTP increases actin polymerization in the dendritic shaft proximal to the uncaging site, we tested whether sLTP-mediated actin polymerization confined GluA1-HT vesicles. By tracking GluA1-HT vesicles after sLTP (*Figure 3D* and *Figure 3—video 1*), we find that sLTP significantly reduces the fraction of GluA1-HT vesicles undergoing active transport, as well as the diffusion coefficient of GluA1-HT vesicles, inside but not outside regions of the dendritic shaft with actin polymerization (*Figure 3E–F*). Moreover, multi-state GluA1-HT vesicles undergoing active transport inside, but not outside, regions of actin polymerization during sLTP have an increased probability of switching to diffusion (*Figure 3—figure supplement 4*). By contrast, sLTP stimulation has no effect on the motion of vesicles in cultures with no MNI (*Figure 3—figure supplement 3C–D*). Importantly, the effect of sLTP on GluA1-HT vesicle motion is disrupted by LatA treatment (*Figure 3G–H* and *Figure 3—figure supplement 4*). Similarly, LatA prevented cLTP-mediated changes in GluA1-HT vesicle mobility, demonstrating that cLTP-induced actin polymerization results in the confinement of GluA1-HT vesicles as well (*Figure 3—figure supplement 5* and *Figure 1—figure supplement 11*).

These results demonstrate that sLTP-mediated actin polymerization in the dendritic shaft confines GluA1-HT vesicles near the site of stimulation, but it is unclear whether confinement actually generates an increased number of vesicles near these sites. After adjusting the number of vesicles for photobleaching (Materials and methods), we find a significant increase in the number of GluA1-HT vesicles inside, but not outside, regions of actin polymerization after sLTP (*Figure 3I* and *Figure 3—figure supplement 3E*). Moreover, the increase in GluA1-HT vesicles is blocked by the addition of LatA (*Figure 3J*). Combined, these observations demonstrate that neurons use actin polymerization as a mechanism to confine and increase the number of vesicles near the sites of synaptic activity.

Having established actin polymerization as the mechanism that mediates stimulation-dependent GluA1-HT vesicle confinement, we sought to determine the mechanism by which F-actin perturbed vesicle motion. AMPARs interact with myosins Va, Vb, and VI (*Correia et al., 2008*; *Wang et al., 2008*; *Nash et al., 2010*; *EstevesdaSilva et al., 2015*), which are involved in the calcium-dependent, short-range recruitment of AMPARs in endosomes to and from dendritic spines. To test if interactions between GluA1 and myosin V and/or VI anchor GluA1 vesicles to F-actin during neuronal activity, we inhibited myosin Va, Vb, and VI by expressing dominant-negative c-terminal domains of these proteins, and by using a pharmacological inhibitor cocktail (MI), during cLTP stimulation (*Figure 3—figure supplement 6A–C*, *Figure 3—video 2* and *Figure 3—video 3*). Inhibition of myosin did not alter the fractions of GluA1-HT vesicles exhibiting active transport, or the diffusion coefficients of GluA1-HT vesicles, either under unstimulated conditions or during cLTP (*Figure 3—figure supplement 6B–C*). Likewise, acute pharmacological inhibition of myosin did not affect the fraction of GluA1-HT vesicles

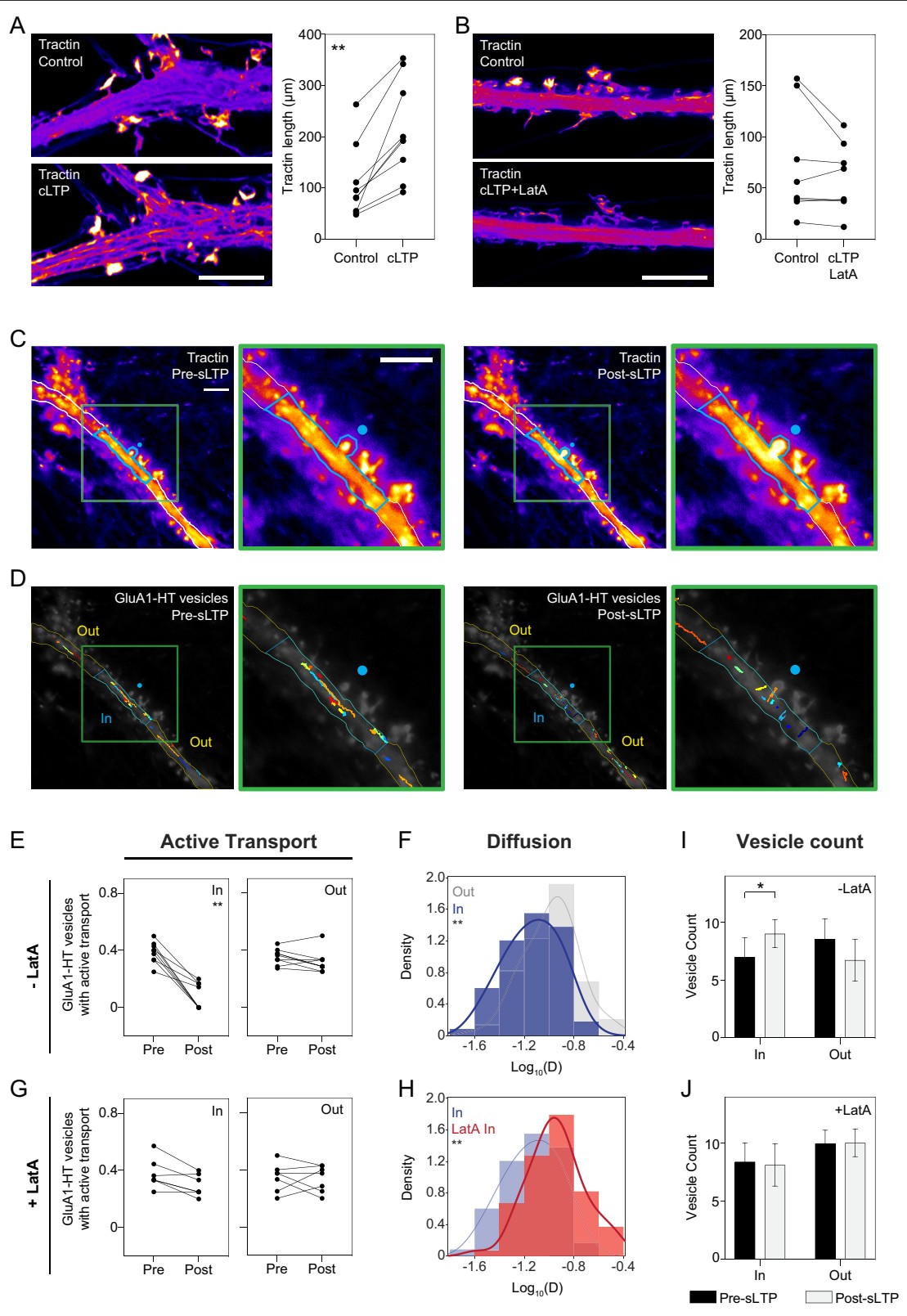

**Figure 3.** Reduced motion of GluA1-HT vesicles during synaptic activity is mediated by actin polymerization in the dendritic shaft. (**A**) Images: representative Airyscan images of F-tractin-mNeongreen (Tractin) in a dendrite before treatment (Control) and during cLTP (cLTP). Scale bar, 5 μm. Graph: combined length of tractin filaments (Tractin length) before treatment and during cLTP. **p=0.0039 by Wilcoxon matched pairs test. Each dot represents the tractin length in an imaged region of a dendrite. n=9 dendrite regions (each from one neuronal culture). (**B**) Same as (**A**), but in

*Figure 3 continued on next page*

*Figure 3 continued*

the presence of Latrunculin A (LatA). Scale bar, 5 µm. n=8 dendrite regions. (**C**) Representative epifluorescence images of tractin immediately before sLTP (Pre-sLTP) and after sLTP (Post-sLTP). Blue dot represents the site of uncaging. Blue outline denotes area with increased tractin signal after sLTP stimulation. Green inset: magnified image of tractin around the uncaging site. Scale bars, 10 µm. (**D**) GluA1-HT vesicle trajectories immediately before sLTP (Pre-sLTP) and after sLTP (Post-sLTP) inside (IN) and outside (OUT) the region where there was increased actin polymerization after sLTP stimulation. Green inset: magnified image of trajectories around the uncaging site. For video, see *Figure 3—video 1*. (**E**) Line graphs of fractions of GluA1-HT vesicles exhibiting active transport inside (IN) and outside (OUT) regions of actin polymerization pre- and post-sLTP. **p=0.0039 by Wilcoxon matched pairs test. Each dot represents the fraction of vesicles exhibiting active transport inside or outside the region with actin polymerization for a single sLTP stimulation experiment. n=9 sLTP stimulation experiments (each experiment targets one spine on one neuron) where GluA1-HT vesicle motion in the dendrite is captured immediately before and after sLTP stimulation. (**F**) Distributions of diffusion coefficients for GluA1-HT vesicles in regions with actin polymerization (IN; blue) versus regions without actin polymerization (OUT; gray) after sLTP stimulation. Lines represent the probability density function of each histogram estimated by kernel density estimation (KDE). **p=0.0053 by Kolmogorov-Smirnov test. n=59–73 GluA1-HT vesicle trajectories pooled together from nine sLTP stimulation experiments. (**G**) Same as (**E**), but in the presence of LatA. IN region defined as the 30 µm region flanking the uncaging site (the average length of dendrite where actin polymerization occurs after sLTP; *Figure 3—figure supplement 3A*), as we do not detect actin polymerization in the presence of LatA. n=7 sLTP stimulation experiments. (**H**) Distributions of diffusion coefficients for GluA1-HT vesicles in regions with actin polymerization (IN; blue) versus GluA1-HT vesicles in similar sized regions in the presence of LatA (LatA IN; red). **p=0.0013 by Kolmogorov-Smirnov test. n=59–67 GluA1-HT vesicle trajectories pooled together from seven to nine sLTP stimulation experiments for each condition. (**I–J**) Bar graphs of adjusted vesicle counts inside (IN) or outside (OUT) regions with actin polymerization after sLTP (**I**) or sLTP in the presence of LatA (**J**). Error bars represent standard deviation. *p=0.0111 by Mann-Whitney test. n=7–9 sLTP stimulation experiments for each condition.

The online version of this article includes the following video, source data, and figure supplement(s) for figure 3:

**Source data 1.** Related to *Figure 3*.

**Figure supplement 1.** Measuring the length of tractin filaments.

**Figure supplement 2.** cLTP stimulation leads to increased actin polymerization.

**Figure supplement 2—source data 1.** Related to *Figure 3—figure supplement 2*.

**Figure supplement 3.** sLTP stimulation increases actin polymerization in the dendritic shaft proximal to the site of stimulation.

**Figure supplement 3—source data 1.** Related to *Figure 3—figure supplement 3*.

**Figure supplement 4.** sLTP-induced actin polymerization increases the probability GluA1-HT vesicles switch from active transport to diffusion.

**Figure supplement 4—source data 1.** Related to *Figure 3—figure supplement 4*.

**Figure supplement 5.** Blocking actin polymerization with LatA disrupts the effect of cLTP on the motion of GluA1-HT vesicles.

**Figure supplement 5—source data 1.** Related to *Figure 3—figure supplement 5*.

**Figure supplement 6.** Inhibition of myosin V and VI does not affect the motion states of GluA1-HT vesicles.

**Figure supplement 6—source data 1.** Related to *Figure 3—figure supplement 6*.

**Figure 3—video 1.** Timelapse sequences of GluA1-HT-JF$_{549}$ after block-and-chase labeling with JF$_{646}$-HTL and JF$_{549}$-HTL pre-sLTP (Figure 3—video 1, Pre-sLTP, left) and post-sLTP (Figure 3—video 1, Post-sLTP, right).

https://elifesciences.org/articles/80622/figures#fig3video1

**Figure 3—video 2.** Timelapse sequence of GluA1-HT-JF$_{549}$ after block-and-chase labeling with JF$_{646}$-HTL and JF$_{549}$-HTL in a dendrite that has been treated with a myosin inhibitor cocktail with no cLTP stimulation (Con +MI).

https://elifesciences.org/articles/80622/figures#fig3video2

**Figure 3—video 3.** Timelapse sequence of GluA1-HT-JF$_{549}$ after block-and-chase labeling with JF$_{646}$-HTL and JF$_{549}$-HTL in a dendrite that has been treated with a myosin inhibitor cocktail during cLTP induction (cLTP +MI).

https://elifesciences.org/articles/80622/figures#fig3video3

exhibiting active transport, or the diffusion coefficient of GluA1-HT vesicles, after sLTP stimulation (*Figure 3—figure supplement 6D*). Based on these observations, we conclude that F-actin disrupts GluA1-HT vesicle motion in a manner independent of myosin activity.

Previous studies have demonstrated that F-actin can constrain the motion of lysosomes (*van Bommel et al., 2019*), leading us to speculate that actin polymerization itself could block the motion of GluA1-HT vesicles. Treatment of neurons with Jasplakinolide (Jsp), an F-actin stabilizer that promotes actin polymerization, is sufficient to inhibit active transport and reduce the rate of diffusion of GluA1-HT vesicles (*Figure 4A* and *Figure 4B*, left histogram). Furthermore, the Jsp-mediated reduction in GluA1-HT vesicle motion is not altered by pharmacological inhibition of myosin, indicating that F-actin itself plays a role in the reduced motion of GluA1-HT vesicles (*Figure 4A* and *Figure 4B*, right histogram).

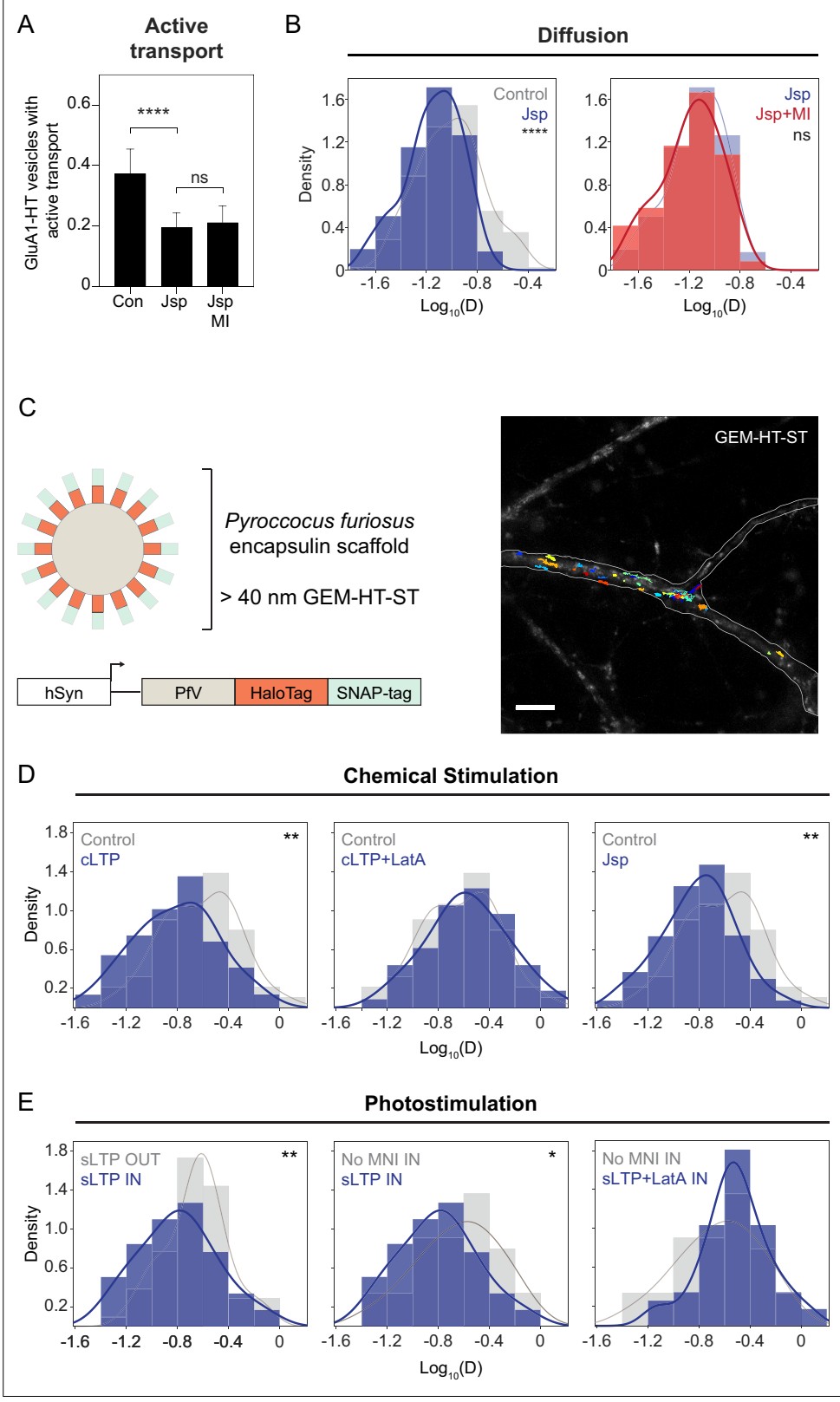

**Figure 4.** F-actin-induced molecular crowding inhibits the motion of particles near the sites of synaptic activity. (**A**) Fractions of GluA1-HT vesicles exhibiting active transport in dendritic shafts without treatment (Con) versus during treatment with Jasplakinolide (Jsp) or Jsp with pharmacological inhibition of myosin (Jsp +MI). Bars represent mean and standard deviation. Con vs Jsp, ****p<0.0001. Significance was determined by Mann-Whitney

*Figure 4 continued on next page*

*Figure 4 continued*

test. n=9–12 timelapse imaging sequences (each timelapse captures the motion of GluA1-HT vesicles in one region of dendrite in one neuronal culture) for each condition. (**B**) Left: distributions of diffusion coefficients of GluA1-HT vesicles in dendritic shafts without treatment (Control; gray) versus during treatment with Jsp (Jsp; blue). Lines represent the probability density function of each histogram estimated by kernel density estimation (KDE). Control vs Jsp, ****p<0.0001 by Kolmogorov-Smirnov test. Right: distributions of diffusion coefficients of GluA1-HT vesicles in dendritic shafts during treatment with Jsp (Jsp; blue) versus Jsp with pharmacological inhibition of myosin (Jsp +MI; red). n=60–227 GluA1-HT vesicle trajectories pooled from 9 to 12 timelapse imaging sequences for each condition. (**C**) Schematic: GEM-HT-ST rheological probe (top). GEM-HT-ST was created by fusing HaloTag (HT) and SNAP-tag (ST) to the PfV protein from *Pyroccocus furiosus* (bottom; *Delarue et al., 2018*). When expressed, PfV-HT-ST fusion proteins self-assemble into a 40 nm encapsulin scaffold that can be labeled with either JF$_{549}$-HTL or JF$_{549}$-SNAP-tag ligand (JF$_{549}$-STL). Image: trajectories of GEM-HT-ST labeled with JF$_{549}$-HTL. Scale bar, 10 µm. For video, see *Figure 4—video 1*. (**D**) Distributions of diffusion coefficients of GEM-HT-ST after chemical stimulation. Control (gray) vs cLTP (blue), **p=0.0047; Control (gray) vs Jsp (blue), **p=0.0011. Significance was determined by Kolmogorov-Smirnov test. n=57–94 GEM-HT-ST trajectories pooled together from six to nine timelapse imaging sequences for each condition. (**E**) Distributions of diffusion coefficients of GEM-HT-ST after sLTP in regions where actin polymerization occurred versus controls. IN region was determined by isolating a 30 µm region flanking the uncaging site (the average length of dendrite where actin polymerization occurs after sLTP; *Figure 3—figure supplement 3A*). sLTP IN (blue) vs sLTP OUT (gray), **p=0.0083; sLTP IN (blue) vs No MNI IN (gray), *p=0.0446. n=44–59 GEM-HT-ST trajectories pooled together from six to nine sLTP stimulation experiments (each experiment targets one spine on one neuron) where GEM-HT-ST motion in the dendrite is captured immediately after sLTP stimulation.

The online version of this article includes the following video and source data for figure 4:

**Source data 1.** Related to *Figure 4*.

**Figure 4—video 1.** Timelapse sequence of GEM-HT-ST labeled with JF$_{549}$-HTL in the dendrite of a cultured rat hippocampal neuron under control conditions.

https://elifesciences.org/articles/80622/figures#fig4video1

---

Previous studies have found that actin can induce molecular crowding in biological systems such as neuronal axons and prevent active transport of vesicles and organelles (*Sood et al., 2018*). To test if actin polymerization could disrupt vesicle motion by changing the rheological properties of the dendritic cytoplasm, we evaluated whether cLTP and sLTP can alter the diffusion of a rheological probe consisting of a genetically encoded multimeric nanoparticle (GEM; *Delarue et al., 2018*; *Figure 4C*). GEM is based on the encapsulin protein PfV of *Pyroccocus furiosus* (*Figure 4C*, schematic), which self-assembles into an icosahedral scaffold of 120 monomers whose size is more similar to GluA1-HT vesicles than are soluble fluorescent proteins. When fused to both HaloTag and the self-labeling SNAP-tag (ST), the particle arising from expression of GEM-HT-ST can be labeled with Janelia Fluor dye ligands and tracked in dendrites (*Figure 4C*, image, and *Figure 4—video 1*).

We find that GEM-HT-ST diffusion is significantly reduced during cLTP induction (*Figure 4D*, cLTP), and this reduction is dependent on actin polymerization as LatA prevents the decrease in diffusion coefficient (*Figure 4D*, cLTP +LatA). Moreover, actin polymerization stimulated by Jsp is sufficient to reduce the motion of GEM-HT-ST (*Figure 4D*, Jsp). sLTP also results in a reduction in the rate of GEM-HT-ST diffusion in regions of dendritic shafts where actin polymerization occurs (*Figure 4E*, sLTP IN vs sLTP OUT) in a manner that is dependent on the presence of MNI (*Figure 4E*, No MNI IN) and actin polymerization (*Figure 4E*, sLTP +LatA IN). These experiments demonstrate that synaptic activity changes the rheological properties of the dendritic shaft by stimulating actin polymerization. Combined with our findings that triggering actin polymerization is sufficient to disrupt GluA1-HT vesicle motion (via Jsp treatment) and that myosin inhibition did not alter activity-mediated changes in motion, these observations are consistent with the hypothesis that actin polymerization confines GluA1-HT vesicles by altering the properties of the dendritic cytoplasm independent of direct interactions between GluA1 and myosin.

## Local increase in GluA1 exocytosis triggered by synaptic activity is dependent on actin-mediated GluA1 vesicle confinement and myosin activity

Although we have shown that actin polymerization can concentrate GluA1 vesicles near the sites of synaptic activity, it is unclear whether this mechanism contributes to increased trafficking of GluA1 to synapses during activity – whether positioning vesicles near the sites of activity also results in increased exocytosis of GluA1-HT at these sites. To study the exocytic rates of endogenous GluA1, we fused super ecliptic pHluorin (SEP) to HaloTag and knocked this tandem reporter into the NTD of GluA1 such that the tag is exposed to the low pH lumen of vesicles during transport (*Figure 5A* and *Figure 5—figure supplements 1–2*). Intracellular GluA1-HT-SEP has low fluorescence until exocytosis, at which point it is exposed to the neutral pH extracellular medium and exhibits strong fluorescence (*Figure 5A*, diagram). During exocytosis, GluA1-HT-SEP released from vesicles is temporarily confined at the sites of exocytosis and appears as bright puncta under fluorescence microscopy (*Figure 5A*, images on right, and *Figure 5—video 1*). GluA1-HT-SEP exocytic events occur at a low rate prior to stimulation, but increase dramatically during cLTP induction (*Figure 5B*, cLTP, and *Figure 5—video 2*), consistent with previous findings (*Kopec et al., 2006*). The increase in GluA1-HT-SEP exocytosis is dependent on actin polymerization, as LatA reduces the rate of exocytosis during cLTP induction (*Figure 5B*, cLTP +LatA). Acute pharmacological inhibition of myosin also disrupts cLTP-stimulated GluA1-HT-SEP exocytosis (*Figure 5B*, cLTP +MI), suggesting that while myosin does not play a role in disrupting transport of GluA1-HT vesicles, it plays a role in regulating exocytosis of GluA1-HT-SEP. These observations demonstrate that actin polymerization and myosin mediate GluA1-HT-SEP exocytosis during cLTP.

We next tested whether sLTP results in local increases in GluA1-HT-SEP exocytosis, and if the increase in exocytosis is spatially correlated with, and dependent on, actin polymerization (*Figure 5C* and *Figure 5—video 3*). In the absence of a marker for F-actin (due to overlapping fluorescence signals between tractin and SEP), we defined the area proximal to the site of stimulation – the 30 µm region surrounding the uncaging site – as the region of actin polymerization based on our previous observations (*Figure 3—figure supplement 3A*). sLTP stimulation increases GluA1-HT-SEP exocytosis events to a much greater extent proximal than distal to the site of uncaging (*Figure 5C*, sLTP). Similar to their effect on cLTP-mediated GluA1-HT-SEP exocytosis, LatA treatment and acute myosin inhibition both partially block sLTP-mediated increases in GluA1-HT-SEP exocytosis (*Figure 5C*, sLTP +LatA and sLTP +MI). When MNI is removed from the media, there is no increase in exocytic events after sLTP (*Figure 5—figure supplement 3*). We conclude that the accumulation of GluA1-HT vesicles near sites of sLTP is spatially correlated with increased exocytosis of GluA1-HT-SEP, and that local disruption of GluA1-HT vesicle motion and the increase in GluA1-HT-SEP exocytosis are both dependent on actin polymerization in the dendritic shaft.

To determine if the concentration of GluA1-HT-SEP vesicles near the site of synaptic activity contributes to increased trafficking of GluA1-HT-SEP to the cell surface, we sparsely labeled GluA1-HT-SEP vesicles with JF dye ligands and simultaneously tracked vesicle motion and exocytosis (*Figure 6A* and *Figure 6—video 1*). We sought to determine whether GluA1-HT-SEP vesicles destined for exocytosis (i.e. pre-exocytosis vesicles) are drawn from a local source or from distal loci via long-range active transport immediately prior to exocytosis. If pre-exocytosis GluA1-HT-SEP vesicles are drawn from local sources, they should travel relatively short net distances to the sites of exocytosis. To test this hypothesis, we measured the radius of confinement (the area in which a trajectory is confined) for the trajectories of pre-exocytosis GluA1-HT-SEP vesicles in unstimulated cells (*Figure 6B*, top bar graph). The trajectories of GluA1-HT-SEP vesicles that undergo exocytosis (Pre-exocytosis) have smaller radii of confinement when compared to vesicles that do not exocytose (Non-exocytosis), demonstrating that GluA1-HT-SEP vesicles are not imported from distal loci immediately prior to exocytosis.

Based on the observations that pre-exocytosis GluA1-HT-SEP vesicles have small search spaces and that stimulation reduces active transport for GluA1-HT vesicles (*Figure 1D–E* and *Figure 2B–C*), we anticipated that stimulation would also reduce active transport for pre-exocytosis GluA1-HT-SEP vesicles, and that these vesicles diffuse over short distances to the sites of exocytosis. To test this idea, we used HMM-Bayes to infer the motion of pre-exocytosis GluA1-HT-SEP vesicles after inducing structural plasticity with sLTP stimulation. Pre-exocytosis GluA1-HT-SEP vesicles exhibit two or more motion states, where the final state before exocytosis is immobility (i.e. vesicles are immobilized by

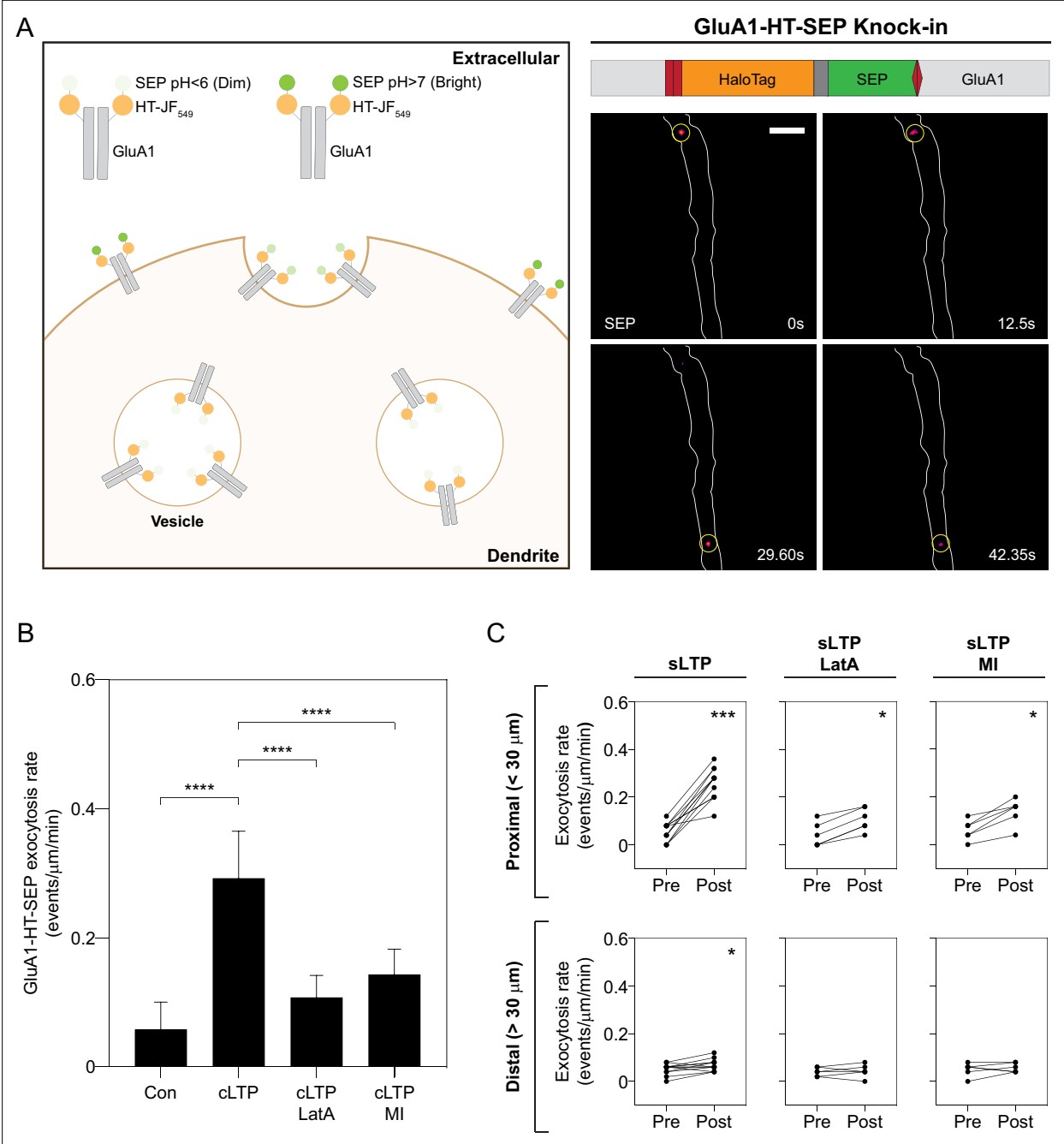

**Figure 5.** Increased GluA1-HT-SEP exocytosis triggered by synaptic activity is dependent on actin polymerization and myosin activity. (**A**) Tagging endogenous GluA1 with a HaloTag-pHluorin tandem fusion reporter (HT-SEP) to track GluA1 exocytosis events. Diagram: endogenous GluA1 tagged with HT-SEP (GluA1-HT-SEP, schematic of fusion on right) has low SEP fluorescence inside vesicles due to the low pH of the vesicle lumen. When a GluA1 exocytosis event occurs, SEP will be exposed to the neutral pH of the extracellular medium, resulting in increased fluorescence. Because receptors are temporarily spatially restricted during exocytosis, a spot with fluorescence can be observed during the event. Images: timelapse of two GluA1-HT-SEP exocytosis events (yellow circles). Scale bar, 5 μm. For video, see *Figure 5—video 1*. (**B**) Bar graph of GluA1-HT-SEP exocytosis in response to cLTP in the absence or presence of Latrunculin A (LatA) or pharmacological myosin inhibition (MI). Bars represent mean events per μm per min and standard deviation. Con vs cLTP, ****p<0.0001; cLTP vs cLTP +LatA, ****p<0.0001; cLTP vs cLTP +MI, ****p<0.0001. Significance was determined by Mann-Whitney test. n=12 timelapse imaging sequences (each timelapse captures GluA1-HT-SEP exocytosis in one region of dendrite in one neuronal culture) for each condition. (**C**) Line graphs of GluA1-HT-SEP exocytic events before and after sLTP (sLTP), sLTP in the presence of LatA (sLTP +LatA), and sLTP in the presence of MI (sLTP +MI). Proximal: GluA1-HT-SEP exocytic events within the 30 μm region surrounding the site of uncaging (i.e. the average length of dendrite where actin polymerization occurs near the site of uncaging after sLTP; see *Figure 3—figure supplement 3A*). sLTP, ***p=0.0005; sLTP +LatA, *p=0.0312; sLTP +MI, *p=0.0312. Distal: GluA1-HT-SEP exocytic events outside of the 30 μm region surrounding the site of uncaging. sLTP,

*Figure 5 continued on next page*

*Figure 5 continued*

*p=0.0430. Significance was determined by Wilcoxon matched pairs test. Each dot represents the number of exocytic events in the indicated region. n=6–12 sLTP stimulation experiments (each experiment targets one spine on one neuron) where GluA1-HT-SEP exocytosis in the dendrite is captured immediately before and after sLTP stimulation.

The online version of this article includes the following video, source data, and figure supplement(s) for figure 5:

**Source data 1.** Related to *Figure 5*.

**Figure supplement 1.** Endogenous GluA1 tagged with HaloTag-SEP reporter via HITI is trafficked to postsynaptic densities.

**Figure supplement 1—source data 1.** Related to *Figure 5—figure supplement 1*.

**Figure supplement 2.** Currents elicited by GluA1-HT-SEP are similar to currents elicited by untagged GluA1.

**Figure supplement 2—source data 1.** Related to *Figure 5—figure supplement 2*.

**Figure supplement 3.** sLTP stimulation in the absence of MNI does not increase GluA1-HT-SEP exocytosis.

**Figure supplement 3—source data 1.** Related to *Figure 5—figure supplement 3*.

**Figure 5—video 1.** Timelapse sequence of GluA1-HT-SEP exocytosis in the dendrite of a cultured rat hippocampal neuron under control conditions.
https://elifesciences.org/articles/80622/figures#fig5video1

**Figure 5—video 2.** Timelapse sequence of GluA1-HT-SEP exocytosis in a dendrite during cLTP induction.
https://elifesciences.org/articles/80622/figures#fig5video2

**Figure 5—video 3.** Timelapse sequence of GluA1-HT-SEP exocytosis in a dendrite pre-sLTP (Figure 5—video 3, Pre-sLTP, left) and post-sLTP (Figure 5—video 3, Post-sLTP, right).
https://elifesciences.org/articles/80622/figures#fig5video3

docking immediately prior to exocytosis; *Figure 6—figure supplement 1A*). Thus, we characterized the motion states of vesicles prior to immobility (docking). Similar to GluA1-HT vesicles, non-exocytosis GluA1-HT-SEP vesicles (i.e. vesicles that do not exocytose) exhibit reduced active transport in response to sLTP (*Figure 6B*, bottom bar graph, No MNI vs sLTP, and *Figure 6—figure supplement 1B*, Non-exocytosis). Surprisingly, sLTP increased the fraction of pre-exocytosis GluA1-HT-SEP vesicles exhibiting active transport (*Figure 6B*, bottom line graph, No MNI vs sLTP, and *Figure 6—figure supplement 1B*, Pre-exocytosis). Similarly, cLTP also increased active transport of pre-exocytosis GluA1-HT-SEP vesicles (*Figure 6—figure supplement 1C*). These results show that synaptic activity stimulates the active transport of GluA1-HT-SEP vesicles to exocytic sites, even though the net distances they travel are short.

As active transport delivers vesicular cargo much faster than diffusion, this finding is consistent with the model that AMPARs need to be rapidly trafficked to exocytic sites during synaptic activity in order to maintain a rapidly accessible membrane-bound reservoir of receptors (*Huganir and Nicoll, 2013*). Nevertheless, it is striking that synaptic activity increases the active transport of pre-exocytosis GluA1-HT-SEP vesicles while simultaneously confining the motion of non-exocytosis GluA1-HT-SEP vesicles. AMPAR-containing endosomes are primarily trafficked along to the length of dendrites by microtubule-based motors (*EstevesdaSilva et al., 2015*; *Setou et al., 2002*; *Hoerndli et al., 2013*), but can also be recruited by myosin Va and Vb to the sites of exocytosis during LTP induction (*Correia et al., 2008*; *Wang et al., 2008*). Thus, we hypothesized that actin polymerization in the dendritic shaft plays a dual role during synaptic activity by: (1) disrupting microtubule-based transport of GluA1 vesicles near synaptic activity (increasing the concentration of GluA1 vesicles); and (2) acting as a substrate for the myosin-based transport of a minority of vesicles to exocytic sites. This may explain the different motion states we observe between pre- and non-exocytosis GluA1-HT-SEP vesicles during stimulation. Interestingly, we primarily observe GluA1-HT vesicles undergoing transport parallel to the length of the dendritic shaft (longitudinal motion) prior to stimulation (*Figure 1D*, control and *Figure 2B*, Pre-sLTP). After stimulation, we find GluA1-HT vesicles that exhibit motion perpendicular to the length of the dendritic shaft (lateral motion; *Figure 1D*, cLTP and *Figure 2B*, Post-sLTP). Consequently, we asked if vesicles that changed their directional bias in response to stimulation were in fact pre-exocytosis vesicles being transported to their exocytic sites by myosin.

To examine this possibility, we first measured the directional bias, described by the angle theta, of pre-exocytosis GluA1-HT-SEP vesicle trajectories in response to sLTP-stimulated synaptic activity (*Figure 6C*). Theta of ~90° indicates that the vesicle is moving parallel to the dendritic shaft (longitudinal motion; *Figure 6C*, top trajectory), whereas theta of ~0° indicates the vesicle is moving

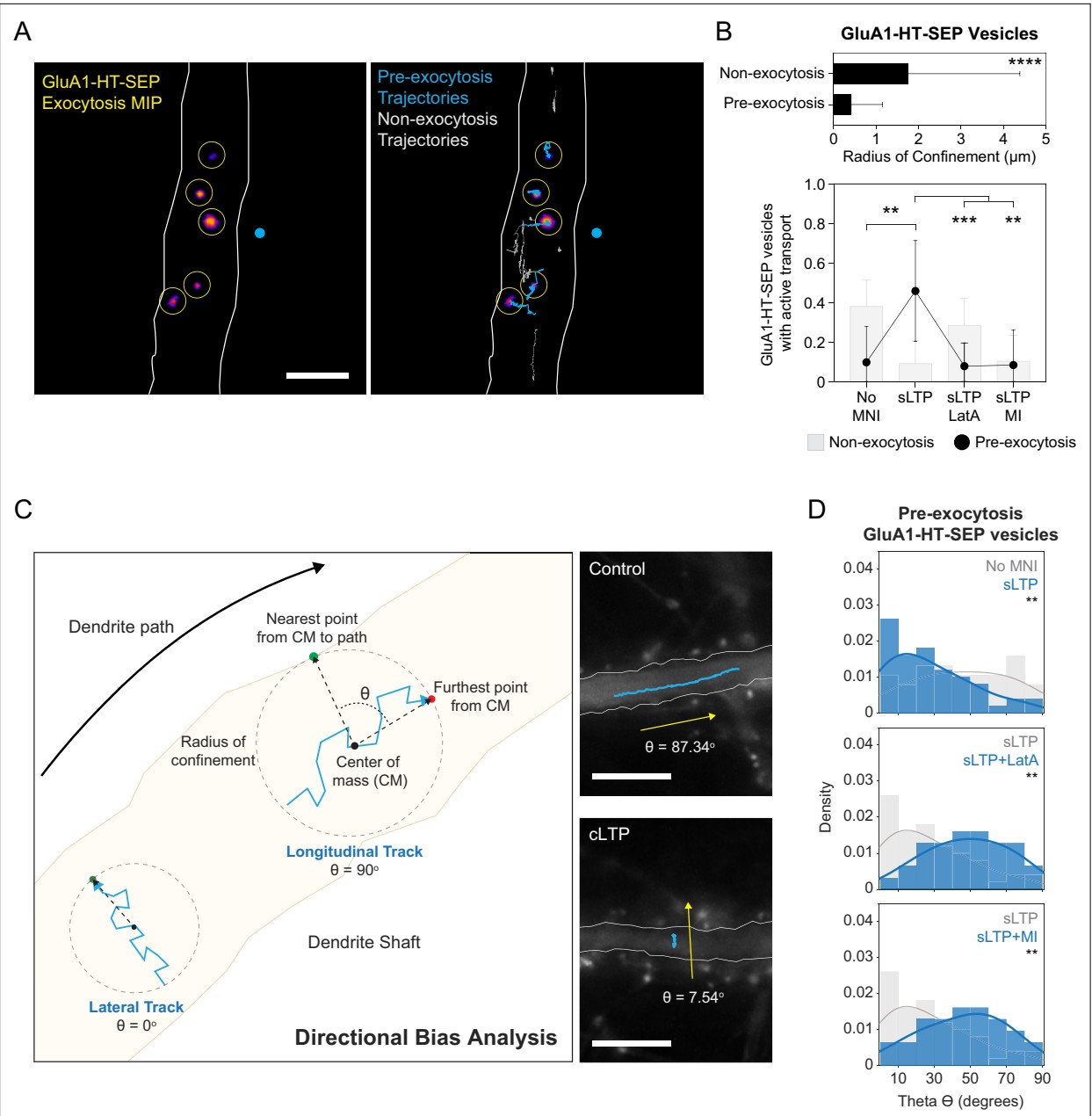

**Figure 6.** GluA1-HT-SEP vesicles are drawn from a local pool in the dendritic shaft prior to exocytosis and exhibit lateral motion that is dependent on actin polymerization and myosin in the shaft during synaptic activity. (**A**) Left: representative maximum intensity projection (MIP) of GluA1-HT-SEP exocytosis over time after sLTP (i.e. the brightest pixels from each image in a timelapse compressed into a single image). Yellow circles indicate exocytic events. Blue dot indicates site of uncaging. Right: GluA1-HT-SEP vesicle trajectories overlaid on the MIP of exocytosis over time. Blue trajectories are GluA1-HT-SEP vesicles that undergo exocytosis (Pre-exocytosis). Gray trajectories are GluA1-HT-SEP vesicles with no detected exocytosis (Non-exocytosis) during the span of imaging. Scale bar, 5 μm. For video. See *Figure 6—video 1*. (**B**) Top: mean radius of confinement for the trajectories of GluA1-HT-SEP vesicles that do not undergo exocytosis (Non-exocytosis) and that do undergo exocytosis (Pre-exocytosis) under unstimulated conditions. Error bars represent standard deviation. ****p<0.0001 by Kolmogorov-Smirnov test. n=65–176 GluA1-HT-SEP vesicle trajectories pooled from five timelapses imaging experiments (each timelapse captures GluA1-HT-SEP motion and exocytosis in one region of dendrite in one neuronal culture). Bottom: bar graph of motion types (*Figure 6—figure supplement 1*) for non-exocytosis GluA1-HT-SEP vesicles (gray bars) and pre-exocytosis GluA1-HT-SEP vesicles (black dots) after sLTP. Only trajectories in regions where actin polymerization occurred were used (see *Figure 3—figure supplement 3A*). No MNI vs sLTP, **p=0.0018; sLTP vs sLTP +LatA, ***p=0.0003; sLTP vs sLTP +MI, **p=0.0017. Significance was determined by Mann-Whitney test. n=9–12 sLTP stimulation experiments (each experiment targets one spine on one neuron) where GluA1-HT-SEP motion and exocytosis in the dendrite are captured immediately after sLTP stimulation. (**C**) Diagram of directional bias analysis. The directional bias of a trajectory can be determined by calculating the angle, theta (Θ), created between the line from the center of mass of a trajectory (CM) to the nearest point on the dendrite path and the

*Figure 6 continued on next page*

*Figure 6 continued*

line from the CM to the furthest point from the CM. Theta of 90° indicates longitudinal movement while theta of 0° indicates lateral movement. Images: representative trajectories of GluA1-HT vesicles traveling longitudinally under control conditions (Control; top) or laterally after cLTP (cLTP; bottom). (**D**) Directional bias for GluA1-HT-SEP vesicles prior to exocytosis (pre-exocytosis) after sLTP. Top: distribution of theta for vesicles after sLTP. sLTP (blue) vs No MNI (gray), **p=0.0067. Middle: distribution of theta for vesicles after sLTP in the presence of an actin polymerization inhibitor (LatA). sLTP +LatA (blue) vs sLTP (gray), **p=0.0032. Bottom: distribution of theta for vesicles after sLTP in the presence of acute myosin inhibition (MI). sLTP +MI (blue) vs sLTP (gray), **p=0.0078. Lines represent the probability density function of each histogram estimated by kernel density estimation (KDE). Significance was determined by Kolmogorov-Smirnov test. n=31–50 GluA1-HT-SEP trajectories pooled from 9 to 12 sLTP stimulation experiments for each condition.

The online version of this article includes the following video, source data, and figure supplement(s) for figure 6:

**Source data 1.** Related to *Figure 6*.

**Figure supplement 1.** Stimulation increases the fractions of pre-exocytosis GluA1-HT-SEP vesicles exhibiting active transport prior to docking.

**Figure supplement 1—source data 1.** Related to *Figure 6—figure supplement 1*.

**Figure supplement 2.** Stimulation changes the directional bias of GluA1-HT vesicles and pre-exocytosis GluA1-HT-SEP vesicles.

**Figure supplement 2—source data 1.** Related to *Figure 6—figure supplement 2*.

**Figure 6—video 1.** Timelapse sequence of GluA1-HT-SEP labeled with JF$_{549}$-HTL (after blocking with JF$_{646}$-HTL) in a dendrite after sLTP stimulation. https://elifesciences.org/articles/80622/figures#fig6video1

---

perpendicular to the dendritic shaft (lateral motion; *Figure 6C*, bottom trajectory). When we examine the directional bias for non-exocytosis GluA1-HT-SEP vesicles in the absence of sLTP stimulation, we find a strong bias for longitudinal motion (*Figure 6—figure supplement 2A*, Non-exocytosis vesicles). By contrast, pre-exocytosis GluA1-HT-SEP vesicles do not exhibit strong biases for either longitudinal or lateral motion in the absence of stimulation (*Figure 6D*, top histogram, No MNI, and *Figure 6—figure supplement 2A*, Pre-exocytosis vesicles), indicating that vesicles travel in random directions to exocytic sites in the absence of synaptic activity. However, when synaptic activity is induced by sLTP, pre-exocytosis GluA1-HT-SEP vesicles have a strong bias for lateral motion (*Figure 6D*, top histogram, sLTP). Likewise, cLTP induction also leads to increased lateral motion for pre-exocytosis GluA1-HT-SEP vesicles (*Figure 6—figure supplement 2B*, Control vs cLTP). These results show vesicles move laterally to exocytic sites in response to synaptic activity, which we speculate is due to vesicles moving from the center to the periphery of the dendritic shaft.

We then sought to determine whether increased lateral motion during synaptic activity is driven by myosin. We first examined the motion states of pre-exocytosis GluA1-HT-SEP vesicles that move laterally during stimulation and find most exhibit active transport (*Figure 6—figure supplement 2C*). This result suggests that GluA1-HT-SEP vesicles move by motor-based transport to exocytic sites. Moreover, when actin polymerization or myosin activity are inhibited, the fraction of pre-exocytosis GluA1-HT-SEP vesicles exhibiting active transport is significantly reduced (*Figure 6B*, bottom line graph, sLTP +LatA and sLTP +MI). Next, we tested the effect of inhibiting actin polymerization on directional bias and find that LatA strongly reduces the lateral motion of pre-exocytosis GluA1-HT-SEP vesicles in response to sLTP (*Figure 6D*, middle histogram, sLTP +LatA). Similarly, pharmacological inhibition of myosin also blocked lateral motion in response to sLTP (*Figure 6D*, bottom histogram, sLTP +MI). Inhibition of either actin polymerization or myosin activity also prevents increased lateral motion of pre-exocytosis vesicles in response to cLTP (*Figure 6—figure supplement 2B*, cLTP vs cLTP +LatA and cLTP +MI). Together, these findings demonstrate that pre-exocytosis GluA1-HT-SEP vesicles are transported laterally by myosin to the sites of exocytosis in response to synaptic activity.

Lastly, we sought to confirm our finding that actin polymerization itself is sufficient to block longitudinal motion. We examined theta for GluA1-HT vesicles in response to stimulation and find a decrease in longitudinal motion and increase in lateral motion (*Figure 6—figure supplement 2D*, sLTP and cLTP), similar to what we observe for pre-exocytosis vesicles. However, when GluA1-HT vesicles are treated with LatA during stimulation, we observe a decrease in lateral motion and a strong increase in longitudinal motion (*Figure 6—figure supplement 2E*, sLTP +LatA and cLTP +LatA). By contrast, inhibition of myosin during stimulation reduces lateral motion but does not increase longitudinal motion (*Figure 6—figure supplement 2F*, sLTP +MI and cLTP +MI, and *Figure 6—figure supplement 2G*). Moreover, inducing actin polymerization with Jasplakinolide (Jsp) while blocking myosin activity reduces longitudinal motion without increasing lateral motion (*Figure 6—figure supplement 2H*, Jsp +MI). These observations support our conclusion that F-actin is necessary and sufficient to

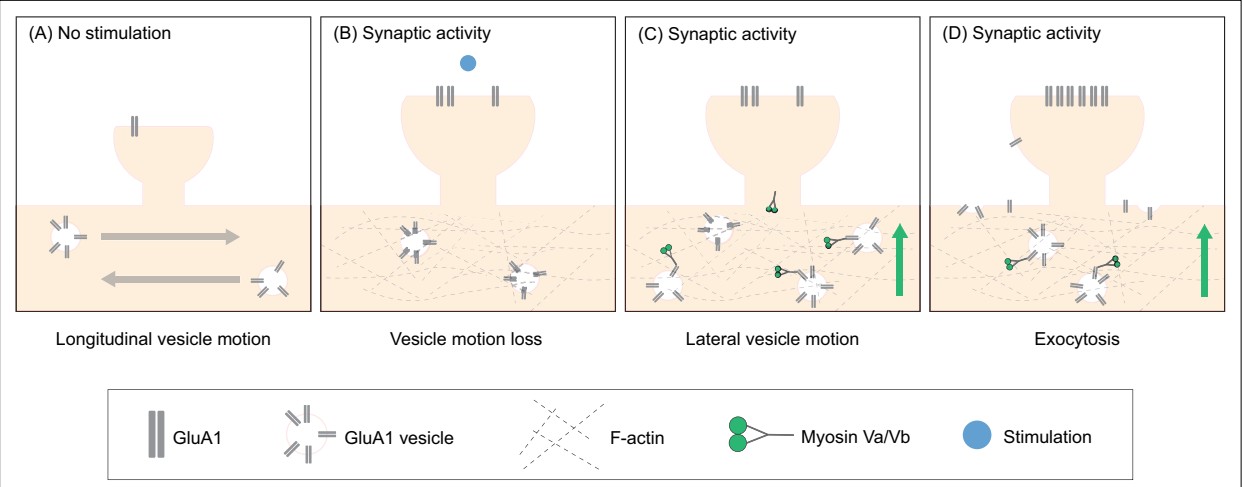

**Figure 7.** Actin polymerization in the dendritic shaft proximal to the site of synaptic activity promotes GluA1 exocytosis by increasing the local pool of GluA1 vesicles and facilitating myosin-dependent transport to the dendrite periphery. (**A**) Under unstimulated conditions, GluA1 vesicles are transported longitudinally along the dendritic shaft. (**B**) When synaptic activity is induced, actin polymerizes in the dendritic shaft near the site of activity. Actin polymerization disrupts the longitudinal motion of vesicles resulting in an increased pool of GluA1 vesicles near the site of activity. (**C**) Myosin V, which is inactive and sequestered in dendritic spines in unstimulated conditions, is activated by calcium influx during synaptic activity. Myosin subsequently translocates to the base of the dendritic spines. (**D**) Myosin V recruits GluA1 vesicles to the periphery resulting in increased exocytosis of GluA1.

confine GluA1 vesicle motion near sites of synaptic activity. Nevertheless, myosin also promotes the surface expression of GluA1 by mediating its active transport to the sites of exocytosis.

## Discussion

In this study, we have developed a novel method to identify, track and characterize the motion of vesicles containing GluA1, enabling us to better understand how AMPARs are delivered specifically to sites undergoing plasticity. We use homology-independent targeted integration (HITI) to tag endogenous GluA1 with HaloTag (GluA1-HT) and then a block-and-chase labeling protocol with Janelia Fluor (JF) dye ligands to achieve a sparse labeling density suitable for the detection of GluA1-HT vesicles. We then utilize single-particle tracking (SPT) followed by hidden Markov modeling with Bayesian model selection (HMM-Bayes) to describe the motion of GluA1-HT vesicles during chemical and structural LTP (cLTP and sLTP). Using this strategy, we find that GluA1-HT vesicles become confined by actin polymerization in the dendritic shaft proximal to sites of stimulation, resulting in an increased intracellular reservoir of GluA1-HT near these sites. Using a pHluorin-HaloTag fusion with GluA1 (GluA1-HT-SEP), we examine how local vesicular reservoirs of GluA1 contribute to GluA1 exocytosis. We find that pre-exocytosis GluA1-HT-SEP vesicles undergo short-range transport perpendicular to the length of the dendritic shaft near sites of stimulation in a manner dependent on both actin polymerization and myosin activity.

Based on these findings, we propose a new model in which neurons utilize actin polymerization in the dendritic shaft to specify the location to which AMPARs are delivered during synaptic activity. First, actin polymerization occurs in the dendritic shaft proximal to the site of synaptic activity, resulting in molecular crowding in the dendritic cytoplasm at this location (*Figure 7A–B*). The increased crowding inhibits the longitudinal motion of GluA1 vesicles, concentrating intracellular GluA1 near the site of synaptic activity (*Figure 7A–B*). F-actin then acts as a substrate for myosin Va and Vb – activated by the influx of calcium (*Correia et al., 2008*; *Wang et al., 2008*) – to recruit GluA1 vesicles to the dendrite membrane (*Figure 7C*). Here, GluA1 vesicles undergo exocytosis, increasing the amount of surface bound GluA1 that can then diffuse into synapses (*Figure 7D*).

Our labeling strategy has key advantages over previously described methods that enable us to image and track GluA1 vesicles. Primarily, we edit genomic copies of *Gria1* to express GluA1-HT and thus avoid pitfalls associated with expressing tagged GluA1 from a plasmid – overexpressing GluA1 can lead to mislocalization and excess formation of calcium permeable GluA1 homomers (*Diering and*

*Huganir, 2018*), altering conductance and neuronal activity. The low level of GluA1-HT expression driven by native *Gria1* promoters is also advantageous because it enables us to achieve sparse particle labeling through a block-and-chase protocol without photobleaching. As photobleaching removes all signal from a designated area, particles are tracked as they travel from outside to inside this region. Consequently, data from photobleaching experiments may be biased towards fast moving particles (i.e. those with active transport) and may omit slowly diffusing particles. HITI can potentially introduce indels into target genes and knock out copies of *Gria1* in some neurons (*Suzuki et al., 2016*), but we find a low frequency of indels around the insertion site of HaloTag (*Figure 1—figure supplement 1D*), and also that *Gria1* expression levels in transfected neurons are similar to those in untransfected neurons (*Figure 1—figure supplement 2B*). These observations indicate that HITI-mediated editing of *Gria1* does not often lead to unwanted mutations, especially those that knock out copies of *Gria1*. Nevertheless, tagging strategies with higher accuracy (e.g. vSlender; *Nishiyama et al., 2017*) could be viable alternatives to HITI.

Our observations of GluA1 vesicle trafficking differ somewhat from previous reports (*Esteves-daSilva et al., 2015*; *Bowen et al., 2017*; *Hangen et al., 2018*). *Hangen et al., 2018* found that GluA1 vesicles undergoing active transport had decreased velocity and paused (i.e. temporarily lost active transport) more frequently during cLTP and photostimulation. By contrast, we find vesicles switch their motion state to diffusion and rarely switch back to active transport (i.e. vesicles stably lose active transport) in response to cLTP and sLTP. Vesicles also exhibit reduced diffusion coefficients after stimulation. Based on these differences, we conclude that GluA1 vesicles are confined, not paused, near sites of synaptic activity. These discrepancies are likely attributable to differences in our methodological approaches. Importantly, our labeling and analysis strategy enabled us to characterize the diffusion coefficient and state-switching probabilities of GluA1-HT vesicles, not just parameters associated with active transport. In addition, we use a specific MNI uncaging protocol to stimulate spine plasticity (not just calcium transients). We find that this protocol also induces actin polymerization in the dendritic shaft, which we demonstrate is the mechanism underlying vesicle confinement.

The precise mechanistic details of how actin polymerization is stimulated in the dendritic shaft and how it regulates GluA1 vesicles during synaptic activity remain to be determined. F-actin is found at the base of dendritic spines (*Schätzle et al., 2018*), and in the dendritic shaft in shaft synapses (*van Bommel et al., 2019*) and in periodic submembrane actin rings (*Lavoie-Cardinal et al., 2020*). Whether these F-actin networks remodel to regulate GluA1 vesicle motion during synaptic activity is not known. Furthermore, it is unclear if F-actin also coordinates the localization of other proteins and organelles near synaptic activity. For example, F-actin patches position lysosomes in shaft synapses to support AMPAR turnover (*Goo et al., 2017*; *van Bommel et al., 2019*), and thus could play a similar role near sites of synaptic activity. However, if actin-induced molecular crowding is generally obstructive to motion, additional mechanisms may be necessary to ensure only relevant particles localize to sites of synaptic activity. For example, myosin cargo adaptors that interact with GluA1 may further enhance the specificity of GluA1 transport (*Correia et al., 2008*; *Hammer and Sellers, 2012*), while other motors might enable GluA1-negative vesicles to bypass F-actin blockades (*Ferro et al., 2019*). Importantly, AMPARs are transported to multiple loci along a single dendrite, suggesting there must also be a mechanism to ensure not all GluA1 vesicles are trapped at a single synapse. One possibility is that there is a sufficient pool of GluA1 vesicles undergoing anterograde and retrograde transport to reach multiple synapses. Further investigation could clarify how actin polymerization is regulated in the dendritic shaft, and how F-actin filters specific cargo in response to synaptic activity.

Previous studies have found that AMPARs are primarily transported along the length of the dendrites on microtubules (*EstevesdaSilva et al., 2015*; *Setou et al., 2002*; *Hoerndli et al., 2013*; *Hoerndli et al., 2015*), but we and others find that short-ranged transport near the site of synaptic activity is mediated by Myosin Va/b (*Correia et al., 2008*; *Wang et al., 2008*). These observations suggest that GluA1 vesicles are transferred from microtubules to actin for local transport. The exact mechanism for this exchange is not known but our findings indicate that F-actin-induced crowding results in the detachment of GluA1 vesicles from microtubules prior to attachment and transport on F-actin. Thus, F-actin enhances local transport by acting as a substrate for myosin-based cargos, but also by creating a cytoplasmic environment that strongly disfavors microtubule-based transport (preventing transport away from the stimulated site). Such a mechanism could have important implications for how cargo

in general is transferred since changes to the subcellular environment could alter modes of transport indirectly rather than through the interactions between motors, cargo, and cytoskeleton.

The precise control of synaptic protein trafficking is vital to synaptic transmission, and thus learning and memory. Nevertheless, many mechanisms regulating synaptic protein trafficking remain to be fully understood. Here, we identify a novel mechanism through which actin polymerization in the dendritic shaft can regulate the surface expression of GluA1 specifically at sites with stimulating inputs. Because F-actin can exert direct and indirect effects on a variety of particles in the dendritic cytoplasm, our findings raise interesting questions regarding whether actin polymerization is a general mechanism to coordinate the delivery of proteins during synaptic plasticity. Elucidating whether and how actin regulates the motion of proteins in the dendritic shaft during neuronal activity could help us better understand the cellular bases for learning and memory.

## Materials and methods

Animal work was conducted according to the Institutional Animal Care and Use Committee guidelines of Janelia Research Campus (IACUC protocol #21–0206).

### Plasmid construction

To generate the HaloTag donor construct (px552-sg-Gria1-HT), miRFP670 was first amplified from pBAD/His-miRFP670 (*Shcherbakova et al., 2016*) and cloned into the KpnI and EcoRI sites of PX552 (*Swiech et al., 2015*). HaloTag was then amplified from LZ10 PBREBAC-H2BHalo (*Li et al., 2016*) using primers that add glycine-serine linkers and *Gria1* sequences to be targeted by Cas9. This HaloTag amplicon was cloned into the XbaI site of PX552 to generate px552-Gria1-HT. To introduce the single guide RNA (sgRNA) insert into px552-Gria1-HT for Cas9 to target *Gria1*, we ordered the 20 bp *Gria1* sequence to be targeted by Cas9 with 5' overhangs (ACC and AAC) from Integrated DNA Technologies (IDT, Newark, NJ), and cloned the target sequence into the SapI sites of px552-Gria1-HT (as previously described in *Swiech et al., 2015*), creating px552-sg-Gria1-HT. For the HaloTag-SEP donor construct (px552-sg-Gria1-HT-SEP), HaloTag was amplified from LZ10 PBREBAC-H2BHalo and SEP was amplified from pAS1NB c Rosella I (*Rosado et al., 2008*). We used primers that add glycine-serine linkers and *Gria1* sequences to be targeted by Cas9 to the 5'-end of HaloTag and the 3'-end of SEP, and primers that introduce a short glycine-serine linker between HaloTag and SEP. The PCR amplicons for HaloTag and SEP were fused using overlap extension PCR, and cloned into the MfeI and NheI sites of px552-sg-Gria1-HT to generate px552-sg-Gria1-HT-SEP. The Cas9 expression construct (PX551) was previously described (*Swiech et al., 2015*). To generate the tractin construct (hSyn-tractin), F-tractin-mNeongreen (also known as ITPKA-mNeongreen) was amplified from the ITPKA-mNeongreen construct (*Chertkova et al., 2017*) and cloned into the KpnI and BsrGI sites of px552-Gria1-HT. Additional sequences between the inverted terminal repeats (ITRs) that are not related to the expression of F-tractin-mNeongreen (i.e. the sgRNA insert and the HaloTag donor sequence) were removed by digesting the construct with MluI and NheI and inserting a short double-stranded DNA oligo with 5' overhangs that are complementary to MluI and NheI, creating hSyn-tractin. The GEM-HT-ST construct (hSyn-GEM) was created by amplifying PfV from pCMV-PfV-Sapphire-IRES-DSRed (*Delarue et al., 2018*), fusing PfV, HaloTag, and SNAP-tag by overlap extension PCR, and cloning the product into the KpnI and BsrGI sites of hSyn-tractin, creating hSyn-GEM. To generate a construct to express the dominant-negative c-terminal domain of myosin VI, we purchased a gBlock of human myosin VI corresponding to amino acids 835–1285 (IDT, Newark, NJ; *Correia et al., 2008*) and cloned the gBlock into the BsrGI and EcoRI sites of the pEGFP-C1 vector (pEGFP-MyosinVI-Ctail; Clontech). To generate pCAG-GluA1-P2A-GFP, GluA1 was amplified from pCMV2-SEP-GluA1 (*Blanco-Suarez and Hanley, 2014*) and fused to P2A-GFP by overlap extension PCR. This amplicon was cloned into the SphI and XhoI sites of pCAGGS (*Niwa et al., 1991*). To generate pCAG-GluA1-HT-P2A-GFP, HaloTag was amplified from px552-Gria1-HT and fused to the N-terminal of GluA1 by overlap extension PCR. This amplicon was cloned into the SphI and BstBI sites of pCAG-GluA1-P2A-GFP. To generate pCAG-GluA1-HT-SEP-P2A-mScarlet, HaloTag-SEP was first amplified from px552-sg-Gria1-HT-SEP and fused to the N-terminal of GluA1 by overlap extension PCR. This amplicon was then cloned into the SphI and BstBI sites of pCAG-GluA1-P2A-GFP to generate pCAG-GluA1-HT-SEP-P2A-GFP. Lastly, we replaced

GFP with mScarlet by cloning P2A-mScarlet into the AgeI and XhoI sites of pCAG-GluA1-HT-SEP-P2A-GFP. All plasmids were propagated in Stbl3 *E. coli* cells.

## Cell culture and transfection

Human embryonic kidney (HEK) 293T cells were cultured in Dulbecco's Modified Eagle Medium (DMEM) with 10% fetal bovine serum (FBS) at 37 °C and 5% $CO_2$. At 70% confluence, HEK293T cells were dissociated and transiently transfected in suspension with plasmids expressing rat GluA1 (pCAG-GluA1-P2A-GFP), GluA1-HT (pCAG-GluA1-HT-P2A-GFP), or GluA1-HT-SEP (pCAG-GluA1-HT-SEP-P2A-Scarlet) at 0.4 µg using Lipofectamine 3000. Cells were then plated onto poly-L-lysine functionalized coverslips. After 48 hr of recovery, cells were used for voltage-clamp recordings. Dissociated hippocampal neurons were prepared from P0 Sprague-Dawley rat pups (Charles River). Hippocampi were dissected out and digested with papain in dissection solution (10 mM HEPES in Hanks' balanced salt solution; HBSS). After digestion, the tissues were gently triturated in minimum essential media (MEM) with 10% FBS and filtered with a 40 µm cell strainer. The cell density and viability were determined by labeling neurons with trypan blue and counting cells with a Countess 3 cell counter. To transfect neurons via electroporation, 500,000 neurons were resuspended in 20 µL complete P3 (P3 solution with supplement) and moved to a cuvette containing 0.5 µg of each plasmid to be transfected. Samples were then electroporated using an Amaxa 4D-Nucleofector with settings CU-110. 80 µL of plating media (MEM with 10% fetal bovine serum, 28 mM glucose, 2.4 mM $NaHCO_3$, 100 µg/mL transferrin, 25 µg/mL insulin, 2 mM L-glutamine) was added to the cuvette immediately after electroporation and samples were allowed to recover for 5 min at 37 °C and 5% $CO_2$. The electroporated sample was then removed from the cuvette and added to an Eppendorf tube with 500 µL plating media. Approximately 50,000–75,000 cells were spread onto the poly-D-lysine-coated coverslip of a 10 mm MatTek dish. Six hours later, plating media was replaced by 2 mL NbActiv4 neuronal culture media. Half of the neuronal culture media was removed and replaced with fresh NbActiv4 every week until the neurons were used.

## Adeno-associated virus (AAV) packaging and transduction

px552-sg-Gria1-HT, px552-sg-Gria1-HT-SEP, PX551, hSyn-tractin, hSyn-GEM, hSyn-GCaMP6s, CAG-tdTomato, and hSyn-GFP were packaged into adeno-associated virus (AAV) by the Janelia Research Campus Viral Services Shared Resource. Briefly, HEK293T cells were transiently transfected with 84 µg of DNA at a ratio of pHelper plasmid:capsid plasmid:AAV construct = 3:2:5. Transfected cells were replenished with fresh serum- and phenol-free DMEM at 6–8 hr post-transfection and incubated for 3 days at 37 °C and 5% $CO_2$. AAVs were collected from both cells and supernatant and purified by two rounds of continuous cesium chloride density gradient. AAV preparations were dialyzed, concentrated to 100 µL, and sterilized by filtration. The final viral titers were measured by quantitative PCR (qPCR) on the inverted terminal repeats (ITRs). AAVs were pseudotyped with AAV2/9, SL1 (retro), or rh10 capsids. For AAV2/9-hSyn-tractin, AAV2/9-hSyn-GEM, AAV2/9-hSyn-GCaMP6s, AAV2/9-CAG-tdTomato, or AAV2/9-hSyn-GFP, $1 \times 10^8$ genomic copies of virus was mixed into 50 µL of NbActiv4 and then added directly into neuronal cultures between 3 and 7 days in vitro (DIV3-7). Fluorescence signals from these reporters were detectable 3 days after transduction.

## Homology-independent targeted integration (HITI)

We used homology-independent targeted integration (HITI) to insert HaloTag or HaloTag-SEP into the endogenous loci of *Gria1* (see *Figure 1A* and *Figure 1—figure supplement 1*). px552-sg-Gria1-HT, px552-sg-Gria1-HT-SEP, and PX551 were delivered into cultured neurons by either electroporation or AAV-mediated transduction. For electroporation, 0.5 µg of px552-sg-Gria1-HT/HT-SEP and 0.5 µg PX551 were mixed with neurons suspended in complete P3 and electroporated as described above (see Cell culture and transfection). For AAV-mediated transduction, $1 \times 10^{10}$ genomic copies of rh10-px552-sg-Gria1-HT/HT-SEP virus and $1 \times 10^9$ genomic copies of rh10-PX551 virus were mixed into 50 µL of NbActiv4 and then added directly into neuronal cultures at DIV3. HaloTag and HaloTag-SEP positive cells could be observed 7 days after transduction. To determine the HaloTag knock-in efficiency (HaloTag KI), we first counted the number of neurons expressing HaloTag and the number of neurons expressing the miRFP670 transfection marker (also expressed from px552-sg-Gria1-HT/HT-SEP) in a

dish. We then calculated the knock-in efficiency by dividing the number of HaloTag$^+$ neurons with the number of neurons that have been transfected with both px552-sg-Gria1-HT and PX551:

$$HaloTag\ KI = \frac{\#\ HaloTag^+\ neurons}{\left(\frac{\#\ miRFP670^+\ neurons}{\#\ total\ neurons}\right)^2 \times\ \#\ total\ neurons}$$

Because PX551 has no transfection marker, we assumed that the rate of PX551 transfection is equal to the rate of px552-sg-Gria1-HT transfection. Consequently, we extrapolate that the rate of co-transfection is equal to the rate of px552-sg-Gria1-HT transfection squared.

## Sequencing

To determine the rate of indels in *Gria1* caused by HaloTag insertion, we first extracted genomic DNA from transfected neurons using a DNEasy Blood and Tissue Kit (Qiagen). The insertion sites of HaloTag were amplified using primers that flanked either the 5' or 3' end of HaloTag. For the 5' end, the forward primer sits 123 bp upstream of HaloTag in Exon 6, while the reverse primer sits 442 bp downstream of the insertion site inside HaloTag. For the 3' end, the forward primer sits 513 bp upstream of the insertion site inside HaloTag, while the reverse primer sits 93 bp downstream of the insertion site in Exon 6. Amplicons were then cloned into pUC vectors and transformed into *E. coli*. Single colonies were selected and the amplicons flanking the insertion sites were sent to GENEWIZ (Azenta Life Sciences) for Sanger sequencing. To identify examples in which HaloTag was inserted in the incorrect orientation, we reversed the HaloTag primers. While we could amplify mock plasmid DNA with HaloTag fused to *Gria1* in the incorrect orientation, we failed to amplify genomic DNA, indicating that the majority of HaloTag insertion are in the correct orientation.

## Janelia Fluor (JF) dye labeling

Janelia Fluor 549 HaloTag ligand (JF$_{549}$-HTL), Janelia Fluor 646 HaloTag ligand (JF$_{646}$-HTL), and cell-membrane impermeable Janelia Fluor 549 HaloTag ligand (JF$_{549}$i-HTL) were kind gifts from Dr. Luke Lavis. Dyes were reconstituted in DMSO and diluted to working concentrations of 10 nM (JF$_{549}$-HTL and JF$_{549}$i-HTL) or 20 nM (JF$_{646}$-HTL) in imaging buffer (150 mM NaCl, 2 mM CaCl$_2$, 2 mM MgCl$_2$, 5 mM KCl, 10 mM HEPES, 30 mM D-glucose, pH 7.4) before use. To label HaloTag expressed in rat hippocampal neurons, NbActiv4 was removed and replaced with imaging buffer containing working concentrations of JF dye-HTL. Neurons were incubated with JF dye-HTL for 30 min at 37 °C and 5% CO$_2$. Cells were then washed three times with imaging buffer and allowed to incubate for another 30 min at 37 °C and 5% CO$_2$. After the second incubation, cells were washed an additional three times with imaging buffer and then used for stimulation or imaging. For block-and-chase labeling experiments, neurons expressing HaloTag were first labeled with 20 nM JF$_{646}$-HTL under 50 nM cycloheximide (CHX) in NbActiv4 for 1 hr at 37 °C and 5% CO$_2$. JF$_{646}$-HTL and CHX were removed by rinsing neurons with NbActiv4 three times. Neurons were then incubated in NbActiv4 for 3 hr at 37 °C and 5% CO$_2$ to allow protein synthesis to recover. After the recovery period, neurons were labeled with 10 nM JF$_{549}$-HTL in imaging buffer for 30 min at 37 °C and 5% CO$_2$. Finally, neurons were rinsed an additional three times with imaging buffer and stimulated or imaged.

## Immunofluorescence

Cultured rat hippocampal neurons were fixed between DIV12 and 21. Neurons were fixed with fixation buffer (1 x PBS, 4% PFA, 4% sucrose, 1 mM MgCl$_2$, 0.1 mM CaCl$_2$) for 20 min. Fixation solution was quenched with 0.1 M glycine in PBS-MC (1xPBS, 1 mM MgCl$_2$, 0.1 mM CaCl$_2$) for 10 min, and neurons were washed three times for 5 min each wash with PBS-MC. Neurons were then simultaneously permeabilized and blocked with 0.1% saponin blocking buffer (0.1% saponin and 5% normal goat serum in PBS-MC) for 1 hr at room temperature. Neurons were labeled with primary antibodies in 0.02% saponin blocking buffer overnight at 4 °C. Neurons were washed three times for 10 min each wash with 0.02% saponin blocking buffer. Neurons were labeled with secondary antibodies in 0.02% saponin blocking buffer for 1 hr at room temperature. Neurons were then washed three times for 10 min each wash with 0.02% saponin blocking buffer and an additional three times for 5 min each wash with PBS. After labeling, neurons were mounted in Prolong Diamond Antifade Mountant for 24 hr. Stimulation emission depletion (STED) microscopy was performed on an Abberior Expert Line

STED microscope equipped with 405/440/485/561/640 nm laser lines for illumination and 595/775 nm laser lines for depletion. Emitted light was collected with four spectral avalanche photodiodes (APDs). A 60 x Plan Apochromat oil-immersion objective (NA = 1.42) was used for all STED imaging. The pinhole size was set to 0.71 airy units (AU). A pixel size of 20 nm was used. 20% depletion laser was used in *xy* and 10% was used in *z*.

## Colocalization analysis

To examine colocalization between HaloTag and GluA1, we performed intensity-based colocalization on antibody-labeled immunofluorescence images using Imaris. STED images were saved as TIFFs and imported to Imaris. To perform colocalization analysis in a specific neuron of interest (e.g. a neuron expressing HaloTag), we used Imaris to create a surface object of the cell of interest and then masked signal outside of the surface object. We then applied an automatic intensity threshold using Otsu's method – an established thresholding method that enables us to minimize subjective analysis – to each individual fluorescence channel to be used for colocalization analysis. Using these threshold levels, we created a new volume channel corresponding to the colocalizing voxels between the two fluorescence channels. We report both the Pearson's coefficient (which indicates whether the intensities of two signals co-vary) and the Manders' coefficients (which indicate whether two signals overlap). For punctate signals that co-occurred inside a spine, but did not colocalize (i.e. HaloTag/GluA1 and postsynaptic density markers), we measured the distance between fluorescence signals using Abberior Imspector. First, images from the two channels (e.g. TRITC for HaloTag and Cy5 for PSD-95) were overlaid and a line profile was applied to adjacent signals (i.e. a single line was drawn through the center of an orange puncta and far-red puncta) in a single spine. We then applied a gaussian fit to each fluorescence intensity profile and measured the distance between the two peaks.

## Hybridization chain reaction RNA fluorescence in situ hybridization (HCR RNA-FISH)

Probes targeting *Rattus norvegicus Gria1* and HaloTag mRNA were designed by Molecular Instruments. Cultured rat hippocampal neurons were fixed as described (see Immunofluorescence). Neurons were permeabilized with 0.1% Triton X-100 in PBS-MC for 15 min. Neurons were washed three times with PBS-MC for 5 min each wash and then two times with 2 x SSC (0.3 M NaCl and 0.03 M sodium citrate) for 5 min each wash. Neurons were next incubated for 30 min at 37 °C in 30% probe hybridization buffer (30% formamide, 5 x SSC, 9 mM citric acid pH 6, 0.1% Tween-20, 50 µg/mL heparin, 1 x Denhardt's solution, 10% low molecular weight dextran sulfate). *Gria1* and HaloTag mRNA were then labeled with *Gria1*- and HaloTag-targeting hybridization probes in 30% probe hybridization buffer for 12 hr at 37 °C. After hybridization, neurons were washed four times with 30% probe wash buffer (30% formamide, 5 x SSC, 9 mM citric acid pH 6, 0.1% Tween-20, 50 µg/mL heparin) and three times with 5 x SSCT (5 x SSC, 0.1% Tween 20) for 5 min each wash at room temperature. Neurons were then incubated with amplification buffer (5 x SSC, 0.1% Tween-20, 10% low molecular weight dextran sulfate) for 30 min at room temperature. Hybridization probes were then amplified with amplification probes conjugated with Alexa 488 or Alexa 546 in amplification buffer for 45 min at room temperature. After amplification, neurons were washed two times with 5 x SSCT for 5 min each, two times with 5 x SSCT for 30 min each, and once with 5 x SSCT for 5 min at room temperature. After HCR RNA-FISH labeling, cells were mounted in Prolong Diamond Antifade Mountant for 24 hr. Confocal images of the labeled samples were taken with an inverted Carl Zeiss LSM 880 microscope equipped with 405/488/561/594/633 nm laser lines for illumination, and 2 multi-Alkali photomultiplier tubes (PMTs) and a 32-channel spectral gallium arsenide phosphide (GaAsP) PMT. QUASAR detection windows were adjusted optimally for each fluorophore. A 63 x Plan Apochromat oil-immersion objective (NA = 1.4) was used for all HCR RNA-FISH imaging. The pinhole size was set to 1 airy unit (AU) based on the longest emission wavelength detected.

## mRNA quantification and correlated localization analysis

Fluorescent puncta representative of individual mRNA were identified and counted using the FISH-Quant program for MATLAB (*Mueller et al., 2013*). First, neurons were identified using JF$_{646}$-HTL labeled GluA1-HT and manually outlined. Overlapping neurons were removed from analysis. A Gaussian kernel filter was applied to HCR RNA-FISH images to enhance spot-like features. A

pre-detection threshold was determined based on the dimmest and brightest pixel on each image and applied to the image. Additional false positives were then removed based on their low quality score. Pre-detected spots were then fit with a 3D Gaussian function. Detected spots were then visually inspected for one neuronal cell body to determine if the detection parameters were adequate, after which the detection parameters were applied to the remaining cells in the image to quantify the number of spots in each cell. We used object-based colocalization to determine whether spots from two channels (i.e. HaloTag mRNA labeling and *Gria1* mRNA labeling) have correlated localization (*Figure 1—figure supplement 2C*). Two-channel images were first deconvolved with Zen Black and then imported to Imaris. mRNA in each channel were detected using the Imaris Spot Detection algorithm with a default quality filter for thresholding. Detected spots were visually inspected. We then examined the colocalization of thresholded spots from the two channels. A distance cutoff (i.e. the maximum distance within which two spots can be considered correlated in their localization) of 0.5 μm was used, as this corresponds to a physical distance of 1470 linearized basepairs (half the maximum distance in which probes from the two channels can be separated if they are both on the same mRNA – i.e. if *Gria1*-HaloTag is synthesized as a single mRNA). We report the percentage of spots in one channel that are within 0.5 μm of spots from the second channel and vice versa as a metric to determine whether the two labels have hybridized to a single mRNA (i.e. *Gria1*-HaloTag).

## Single-particle tracking

GluA1-HT and GluA1-HT-SEP vesicles were imaged with a Nikon Eclipse TiE inverted microscope equipped with 405/488/561/642 nm laser lines, three iXon Ultra 897 electron multiplying charge-coupled device (EMCCD) cameras connected via a tri-cam splitter for simultaneous multicolor acquisition, an automatic total internal reflection fluorescence (TIRF) illuminator, and a perfect focusing system. A Tokai Hit Stage Top Incubator was used to maintain constant environmental conditions of 5% $CO_2$ and 37 °C during imaging. A 100 x TIRF Apochromat oil-immersion objective (NA = 1.49) was used for imaging. To image GluA1-HT and GluA1-HT-SEP vesicles, we employed highly inclined and laminated sheet (HILO) illumination. Specifically, the TIRF illuminator was adjusted to deliver the laser beam with an incident angle smaller than the total internal reflection angle, which generated a highly inclined light sheet centered on the focal plane. Compared with standard epifluorescence illumination, HILO illumination reduces background from out-of-focus excitation. Images were acquired at 20 Hz. The *xy* pixel size for all GluA1-HT and GluA1-HT-SEP single-particle tracking experiments was 160 nm. 2D single-particle tracking was performed with a custom MATLAB program. Single molecule localization (x,y) was obtained through 2D Gaussian fitting and tracking was based on the multiple-target tracing (MTT) algorithm. The localization and tracking parameters in SPT experiments are listed in the *Table 1*. The resulting tracks were manually curated and inspected to ensure tracks were accurately reconstructed.

**Table 1.** Localization and tracking parameters for the MTT program.

| | | |
|---|---|---|
| Localization error | 1E-6 | |
| Deflation loops | 3 | |
| Maximum number competitors | 3 | |
| Maximum diffusion coefficient (μm²/s) | 1 | For GluA1-HT |
| Maximum diffusion coefficient (μm²/s) | 3 | For GEM |

## Hidden Markov modeling with Bayesian model selection (HMM-Bayes) analysis

The diffusion and transport states of individual GluA1-HT and GluA1-HT-SEP trajectories were analyzed with the HMM-Bayes MATLAB program using default parameters. HMM-Bayes classifies each jumping step in a trajectory as either diffusion or active transport and also calculates the diffusion coefficient and velocity of each step. Active transport (AT) is modeled as directed motion (V) with Brownian diffusion (D) according to the equation:

$$< r^2 >= 4D\Delta t + (V\Delta t)^2$$

We defined the maximum number of unique motion states that can be inferred for a trajectory to be 3 (i.e. a trajectory can be assigned 1, 2, or 3 unique states). Diffusion coefficients determined by HMM-Bayes analysis were validated by comparing these values to diffusion coefficients for the same trajectories determined by MSD curve fitting with MSDanalyzer (*Tarantino et al., 2014*) in MATLAB (minimal fitting of $R^2$=0.8; see *Figure 1—figure supplement 5* for more detail). For any multi-state trajectory that exhibited ambiguity in its motion states (i.e. its motion states appeared to differ subjectively from the motion states assigned by HMM-Bayes, and the motion states were predicted by HMM-Bayes with low probability), we segmented the trajectory based on the apparent motion of each segment and reanalyzed each segment with HMM-Bayes and MSDanalyzer.

## Vesicle identification

After reconstructing trajectories for GluA1-HT and GluA1-HT-SEP particles and characterizing their motion, we identified vesicles for further analysis. To identify vesicles, we first removed particles that clearly localized inside spines for the majority of steps in their trajectories. Next, we filtered all particles that exhibited active transport based on HMM-Bayes analysis. For particles that exhibit only diffusive motion, we then filtered out all particles that bleach completely within 100 frames (the maximum number of frames required for a GluA1-HT homotetramer to bleach; *Figure 1—figure supplement 6B*). Lastly, we removed particles that have diffusion coefficients greater than 0.45 $\mu m^2/s$ (i.e. particles on cell surface; *Figure 1—figure supplement 6C*) or less than 0.02 $\mu m^2/s$ (i.e. particles trapped in postsynaptic densities; *Figure 1—figure supplement 7*). To separate vesicles into zones (e.g. in *Figure 2*), we first measured the length of the dendritic shaft using the GFP fill. As images were typically centered on uncaging sites, we separated the length of the dendritic shaft into 3 equal sections based on proximity to the uncaging site. For example, if a dendrite is 90 µm long, Zone 1 would be the 30 µm length of dendrite immediate flanking the uncaging site; Zone 2 would be the two 15 µm lengths of dendrite flanking Zone 1; and so forth for Zone 3.

## Chemical stimulation

To determine whether chemical LTP (cLTP) increased the concentration of GluA1-HT on neuronal surfaces, we washed cultured neurons three times with stimulation buffer (150 mM NaCl, 2 mM $CaCl_2$, 5 mM KCl, 10 mM HEPES, 30 mM D-glucose, 20 mM bicuculline, 1 µM strychnine, pH 7.4) and then stimulated neurons with 0.2 mM glycine in stimulation buffer for 15 min at 37 °C and 5% $CO_2$. Neurons were rinsed twice with wash buffer (150 mM NaCl, 2 mM $CaCl_2$, 2 mM $MgCl_2$, 5 mM KCl, 10 mM HEPES, 30 mM D-glucose, 20 mM bicuculline, 1 µM strychnine, pH 7.4) and then incubated with 25 nM membrane impermeable $JF_{549}$-HTL ($JF_{549}$i-HTL) in wash buffer for 30 min. Cells were then washed three times with wash buffer and fixed as described (see Immunofluorescence). For live-cell GluA1-HT vesicle tracking experiments, the dish was placed onto the stage of a Nikon Eclipse TiE microscope after three washes with stimulation buffer (see Single-particle tracking for imaging setup). After a desired field of view was identified, stimulation buffer was exchanged for 0.2 mM glycine in stimulation buffer. A desired Z plane was identified and imaging commenced immediately. For Jasplakinolide (Jsp) stimulation experiments, neurons were first washed with imaging buffer and placed onto the Nikon Eclipse TiE microscope. After a desired field of view was identified, Jsp in imaging buffer was added to the dish to a final concentration of 0.25 µM. After 10 min of incubation with Jsp, we commenced imaging. For experiments with Latrunculin A (LatA), neurons were pretreated with 1 µM LatA for 10 min. In addition, LatA was added to every buffer for stimulation and imaging. For experiments examining the effect of Nocodazole on the active transport of particles, neurons were pretreated with 500 nM Nocodazole for four hours prior to imaging. 500 nM Nocodazole was also added to imaging buffer during imaging. For myosin V and VI inhibition, a cocktail of 4 µM Pentabromopseudilin (PBP), 10 µM MyoVin-1, and 4 µM of 2,4,6-Triiodophenol (TIP) was added to stimulation buffer during chemical stimulation and photostimulation.

## Photostimulation

For all photostimulation experiments, neurons were washed three times with photostimulation buffer (150 mM NaCl, 2 mM $CaCl_2$, 5 mM KCl, 10 mM HEPES, 30 mM D-glucose, pH 7.4, plus 1 µM Tetrodotoxin; TTX). Photostimulation buffer was then replaced with 2 mM MNI-caged-L-glutamate (MNI) in photostimulation buffer, and samples were placed onto a Nikon Eclipse TiE microscope (see

Single-particle tracking for imaging setup). For all MNI uncaging experiments, we used a 405 nm uncaging laser (20 mW at the fiber end,~1 μm spot diameter; Nikon). To calibrate the strength of the 405 nm uncaging laser, we used GCaMP6s (*Chen et al., 2013*), a genetically encoded calcium indicator, to monitor the influx of calcium in response to MNI uncaging. Neurons expressing GCaMP6s were identified using the FITC channel. The 405 nm uncaging laser was parked 1 μm from the tip of a dendritic spine head and fired once for 0.1 ms to determine whether the laser power was sufficient to evoke calcium influx at the targeted spine, in neighboring spines, and in the dendritic shaft. After performing uncaging once and recording the calcium influx, we moved to another spine on another neuron and repeated uncaging. We varied the laser power between 10 and 100% to determine the laser power that would maximally activate GCaMP6s only in the targeted spine. For glutamate uncaging-evoked sLTP experiments, we used neurons expressing GFP and GluA1-HT (or just GluA1-HT-SEP for exocytosis experiments). We selected dendritic spines to be targeted for glutamate uncaging-evoked sLTP based on their morphology in the FITC channel. We then positioned the 405 nm uncaging laser 1 μm from the tip of the dendritic spine head. sLTP was stimulated by firing the 405 nm uncaging laser at 1 Hz for 50 s with a pulse width of 0.1 ms. After photostimulation, we checked whether the dendritic spine head of interest had expanded in area in the FITC channel. To examine the effect of sLTP on GluA1-HT vesicle motion, we performed timelapse imaging on GluA1-HT vesicles in the TRITC channel (see Single-particle tracking) immediately before and after sLTP stimulation. For experiments using GluA1-HT-SEP, we identified spines for stimulation using the Cy5 channel (i.e. GluA1-HT-JF$_{646}$) due to the loss of the FITC channel to SEP. To account for photobleaching and determine the adjusted number of vesicles after photostimulation, we determined the vesicle counts before and after sLTP stimulation in no MNI controls. We then used the fold loss in vesicles after sLTP stimulation in no MNI controls to correct the number of vesicles after sLTP stimulation across all conditions.

## Tractin imaging

To examine the effect of cLTP on the length of F-actin fibers in the dendritic shaft, cultured rat hippocampal neurons expressing tractin were imaged using an inverted Carl Zeiss LSM 880 microscope (see Immunofluorescence) with a 63 x Plan Apochromat oil-immersion objective (NA = 1.4). To achieve higher resolution, emitted photons were collected with a 32-channel spectral GaAsP PMT for Airyscan processing. As these experiments were performed on live cells, a PeCon Incubator XL enclosure with Lab-Tek S1 heating insert and CO$_2$ lid were used to maintain constant environmental conditions of 5% CO$_2$ and 37 °C during imaging. Prior to imaging, neurons expressing tractin were washed three times with stimulation buffer (see Chemical stimulation) and placed onto the Zeiss LSM 880 stage. A dendrite of interest expressing tractin was identified and a z-stack of the dendrite acquired. After acquisition of the dendrite before cLTP (pre-cLTP), stimulation buffer was exchanged for 0.2 mM glycine in stimulation buffer. After 15 min, a z-stack of the dendrite (post-cLTP) was acquired with identical imaging parameters. Airyscan processing was applied to both the pre- and post-cLTP images. The F-actin skeletons were then quantified as described in *Figure 3—figure supplement 1*. Briefly, we improved signal-to-noise by applying background subtraction and a minimum filter to the raw images. We then masked tractin signals in spines. We used Otsu's method to threshold filaments and generate a binary mask, which we then skeletonized. Finally, we measured the length of the skeleton using the AnalyzeSkeleton ImageJ plugin (*Arganda-Carreras et al., 2010*).

## Phalloidin labeling

Neurons were fixed and permeabilized as described above (see Immunofluorescence), except MgCl$_2$ and CaCl$_2$ were removed from all buffers. Neurons were then blocked for 1 hr in 0.1% Triton X-100 blocking buffer (3% bovine serum albumin and 0.1% Triton X-100 in PBS) at room temperature. F-actin was then labeled with 0.13 μM Abberior Star 635 p conjugated phalloidin (Abberior) in 0.1% Triton X-100 blocking buffer for 1 hr at room temperature. After labeling, neurons were washed once with PBS and mounted in MOWOIL-DABCO. Superresolution images were acquired using an Abberior Expert Line STED microscope (see Immunofluorescence). While STED imaging greatly improved resolution, it also reduced the strength of F-actin signals. Consequently, we could not use the method described above (see Tractin imaging and *Figure 3—figure supplement 1*) to measure the length of F-actin fibers labeled with phalloidin. Instead, we masked phalloidin signal in spines using the

Intermodes method - a method for automatic intensity thresholding - and measured the mean fluorescence intensity of phalloidin inside the dendritic shaft.

## Radius of confinement and directional bias analysis

To determine the directional bias of a trajectory using angle analysis, we implemented a custom MATLAB program. Specifically, for a given trajectory, we defined the radius of confinement as the distance from the centroid of a trajectory to the furthest point in the trajectory to the centroid. Next, we created a mask around the dendrite in which the trajectory is localized and set the dendritic path along the length of the dendrite. We then determined the point on the dendritic path that is closest to the centroid of the trajectory. The angle, theta $\Theta$, created by these three points (the point on the dendritic path closest to the centroid, the centroid itself, and the point on the trajectory furthest from the centroid) indicates whether the motion of the trajectory is biased *towards* the length of the dendritic path or *along* the length of the dendritic path. Theta ~ 0° indicates that the motion of the trajectory is biased perpendicular to the dendritic path (i.e. lateral motion) while theta ~ 90° indicates that the motion of the trajectory is biased parallel to the dendritic path (i.e. longitudinal motion).

## Electrophysiology

Whole-cell voltage-clamp recordings were performed to determine the preservation of GluA1-HT and GluA1-HT-SEP functionality. Cultured hippocampal neurons were assessed at DIV7-8. Labeling GluA1-HT/HT-SEP with JF$_{549}$-HTL was performed 12 hr prior to patch clamp experiments. GluA1-HT/HT-SEP positive and negative neurons were identified via epifluorescence. Whole-cell configuration was achieved using pipettes pulled and polished to a resistance of 2–5 MΩ. SylGard 184 coating was applied to reduce pipette capacitance. Pipette solution composition included 120 mM Cs-MES, 5 mM NaCl, 10 mM TEA-Cl, 5 mM Lidocaine, 1.1 mM EGTA, 10 mM HEPES, 0.3 mM Na-GTP, 4 mM Mg-ATP. External solution contained 135 mM NaCl, 3 mM KCl, 2 mM MgCl$_2$, 1.5 mM CaCl$_2$, 10 mM TEA-Cl, 10 mM HEPES, 10 mM Glucose. The external solution was supplemented with 200 nM TTX and 10 μM AP5. Cell capacitance was estimated and corrected to 80% with a 10 μsec lag. Gap free voltage clamp recordings were performed at 100 kHz acquisition frequency with a 1.0 kHz Bessel filter. Voltage was clamped to –70 mV, correcting for liquid junction potential. Following baseline measurements, 100 μM glutamate in pH adjusted external solution was locally perfused for 5 s using a fused silicate pipe positioned within 100–200 μm of the neuron recorded. The neuron then recovered with continued bath perfusion and local perfusion of glutamate-free external solution. For analysis, the gap-free epochs were reduced 50-fold. The data was normalized to current density via cell capacitance estimations. Peak current density for each response was determined. To determine whether GluA1-HT and GluA1-HT-SEP were functional using a heterologous expression system that does not express GluA1, HEK293T cells were transfected to express GluA1-P2A-GFP, GluA1-HT-P2A-GFP, or GluA1-HT-SEP-P2A-Scarlet (see Cell culture and transfection). After 48 hours of recovery, voltage-clamp recordings of glutamate induced GluA1 currents were performed similarly to the above described neuronal recordings. Patch pipettes pulled to 2–4 MΩ resistance were filled with internal solution containing 2.8 mM KCl, 140 mM NaCl, 2 mM MgCl$_2$, 2 mM CaCl$_2$, 12 mM glucose, and 10 mM HEPES at pH 7.3. After whole-cell access was achieved, a brief baseline recording of external bath solution was performed; the external solution was comprised of 140 mM KCl, 5 mM MgCl$_2$, 5 mM EGTA, and 10 mM HEPES at pH 7.3. Due to the propensity of glutamate contamination in immortalized cell-line cultures, we anticipated desensitization of functionally expressed GluA1. To re-sensitized GluA1, we performed 2 min of constant, local perfusion of external solution supplemented with 50 μM cyclothiazide. After the cyclothiazide pretreatment, 2 mM glutamate with 50 μM cyclothiazide was locally applied to the patched cell for 10 s; this stimulation was followed by a washout period with normal external solution. Currents were adjusted to the control-bath baseline and normalized via cell capacitance. Data is reported as peak glutamate induced current.

## Statistical analysis and experimental replicates

All statistical tests and correlation analysis were performed in GraphPad Prism, version 8. Specific statistical tests to determine the significance of results are indicated in the figure legends. Histograms and probability functions were generated using the Distribution Fitter Application (Statistics and Machine Learning toolbox) in MATLAB. For experiments where GluA1-HT particles were

tracked after chemical stimulation (e.g. cLTP induction), particles were tracked in one segment of a dendrite and then the neuronal culture was discarded. In other words, each timelapse is of a dendrite from an independent neuronal culture. This was done to roughly synchronize the start of each timelapse after chemical stimulation. For all MNI uncaging experiments, one spine was targeted per neuron (i.e. we identified a spine on a new neuron for each sLTP stimulation experiment). A maximum of three spines (i.e. three neurons) were targeted for MNI uncaging in one culture before the culture was discarded. Targeted spines were separated by at least 3 mm to minimize the possibility that a spine was exposed to uncaged MNI from a previous experiment. Typically, two to three neuronal cultures were prepared from a single dissection. Consequently, experimental replicates were performed on neurons derived from at least two rats. Nevertheless, we found no significant difference in either the fraction of GluA1-HT vesicles exhibiting active transport or the mean diffusion coefficient of GluA1-HT vesicles between neurons derived from different dissections under a single condition (*Figure 1—figure supplement 5F–G*). In other words, variation stemming from technical and biological replicates was significantly less than variation due to treatments.

## Materials availability

Newly created materials, including custom MATLAB codes, for this study are available upon request from the corresponding author.

## Acknowledgements

We thank our colleagues Dr. Heejun Choi, Dr. Kyle Harrington, Dr. Jingyi Hou, Dr. Dong-woo Hwang, Dr. Aurelie de Rus Jacquet, Dr. Namsoo Kim, Dr. Xi Long, and Dr. Young J Yoon for useful discussions; Dr. David E Clapham and Dr. Pietro De Camilli for their helpful suggestions and feedback on this manuscript; Dr. Luke D Lavis and Jonathan B Grimm for providing the Janelia Fluor dyes and guidance on their usage; Dr. Damien Alcor from the Janelia Research Campus (JRC) Light Microscopy Center and Dr. Andrian Gutu for their assistance with imaging platforms; Dr. Hyun Ah Yi and Dr. Alina D Gutu from the JRC Viral Tools facility for packaging AAVs; Dr. Kevin McGowan, Sarah Kivimaki, and Jordan Towne from JRC Molecular Genomics for assistance with cloning; Jenny Hagemeier and Leanna Eisenman from JRC Cell Culture Shared Resources for their help with cell cultures; Michelle Quiambao for administrative assistance; and the *eLife* editorial and production teams for assistance in submitting and publishing this manuscript. This work was supported by the Howard Hughes Medical Institute.

## Additional information

### Competing interests

Erin K O'Shea: Erin K O'Shea is President of the Howard Hughes Medical Institute, one of the three founding funders of eLife, and a member of eLife's Board of Directors. The other authors declare that no competing interests exist.

### Funding

| Funder | Grant reference number | Author |
|---|---|---|
| Howard Hughes Medical Institute | | Victor C Wong<br>Patrick R Houlihan<br>Hui Liu<br>Deepika Walpita<br>Michael C DeSantis<br>Zhe Liu<br>Erin K O'Shea |

The funders had no role in study design, data collection and interpretation, or the decision to submit the work for publication.

## Author contributions
Victor C Wong, Conceptualization, Resources, Data curation, Software, Formal analysis, Validation, Investigation, Visualization, Methodology, Writing – original draft, Project administration, Writing – review and editing, Designed and performed all experiments except whole-cell voltage clamp recordings; Patrick R Houlihan, Data curation, Software, Formal analysis, Validation, Investigation, Visualization, Methodology, Writing – original draft, Writing – review and editing, Designed and performed whole-cell voltage clamp experiments; Hui Liu, Deepika Walpita, Resources, Methodology; Michael C DeSantis, Resources, Software, Formal analysis, Writing – review and editing; Zhe Liu, Conceptualization, Resources, Software, Formal analysis, Methodology, Writing – review and editing; Erin K O'Shea, Conceptualization, Resources, Supervision, Funding acquisition, Visualization, Project administration, Writing – review and editing

## Author ORCIDs
Victor C Wong ⓘ https://orcid.org/0000-0002-7060-960X
Patrick R Houlihan ⓘ https://orcid.org/0000-0002-2505-2347
Hui Liu ⓘ https://orcid.org/0000-0002-4105-8570
Michael C DeSantis ⓘ https://orcid.org/0000-0002-7214-2740
Zhe Liu ⓘ https://orcid.org/0000-0002-3592-3150
Erin K O'Shea ⓘ https://orcid.org/0000-0002-2649-1018

## Ethics
Animal work was conducted according to the Institutional Animal Care and Use Committee guidelines of Janelia Research Campus (IACUC protocol #21-0206).

## Decision letter and Author response
Decision letter https://doi.org/10.7554/eLife.80622.sa1
Author response https://doi.org/10.7554/eLife.80622.sa2

---

# Additional files

## Supplementary files
• MDAR checklist

## Data availability
All data generated or analyzed during this study are included in the manuscript and supporting files, source data files have been provided for all figures.

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

# Appendix 1

## Appendix 1—key resources table

| Reagent type (species) or resource | Designation | Source or reference | Identifiers | Additional information |
|---|---|---|---|---|
| Antibody | anti-AMPA receptor 1 (GluA1; Extracellular; Guinea pig polyclonal) | Alomone Labs | AGP-009 | IF(1:250) |
| Antibody | anti-Chicken IgY, Alexa Fluor Plus 647 (Goat polyclonal) | Thermo Fisher Scientific | A-32933 | IF(1:500) |
| Antibody | anti-Chicken IgY, Star Orange (Goat polyclonal) | abberior GmbH | STORANGE-1005–500 UG | IF(1:500) |
| Antibody | anti-Chicken IgY, Star Red (Goat polyclonal) | abberior GmbH | STRED-1005–500 UG | IF(1:500) |
| Antibody | anti-GluA1 (Mouse monoclonal) | Synaptic Systems | 182 011 | IF(1:500) |
| Antibody | anti-GluN1 (Mouse monoclonal) | Synaptic Systems | 114 011 | IF(1:1000) |
| Antibody | anti-Guinea Pig, Alexa Fluor 488 (Goat polyclonal) | Thermo Fisher Scientific | A-11073 | IF(1:500) |
| Antibody | anti-Guinea Pig IgG, Alexa Fluor 555 (Goat polyclonal) | Thermo Fisher Scientific | A-21435 | IF(1:500) |
| Antibody | anti-Guinea Pig IgG, Alexa Fluor 594 (Goat polyclonal) | Thermo Fisher Scientific | A-11076 | IF(1:500) |
| Antibody | anti-Guinea Pig IgG, Alexa Fluor 647 (Goat polyclonal) | Thermo FIsher Scientific | A-21450 | IF(1:500) |
| Antibody | anti-HaloTag (Rabbit polyclonal) | Promega | G9281 | IF(1:500) |
| Antibody | anti-Homer1 (Chicken polyclonal) | Synaptic Systems | 160 006 | IF(1:1000) |
| Antibody | anti-Mouse IgG, Alexa Fluor 488 (Goat polyclonal) | Thermo Fisher Scientific | A-11001 | IF(1:500) |
| Antibody | anti-Mouse IgG, Alexa Fluor 546 (Goat polyclonal) | Thermo Fisher Scientific | A-11030 | IF(1:500) |
| Antibody | anti-Mouse IgG, Alexa Fluor 594 (Goat polyclonal) | Thermo Fisher Scientific | A-32742 | IF(1:500) |
| Antibody | anti-Mouse IgG, Alexa Fluor 647 (Goat polyclonal) | Thermo Fisher Scientific | A-21236 | IF(1:500) |
| Antibody | anti-Mouse IgG, Atto 647 N (Goat polyclonal) | Rockland Immunochemicals Inc. | 610-156-121 | IF(1:500) |
| Antibody | anti-PSD-95 (7E3-1B8; Mouse monoclonal) | Thermo Fisher Scientific | MA1-046 | IF(1:1000) |
| Antibody | anti-PSD-95 (Guinea pig polyclonal) | Synaptic Systems | 124 014 | IF(1:2000) |
| Antibody | anti-Rabbit IgG, Alexa Fluor 488 (Goat polyclonal) | Thermo Fisher Scientific | A-11034 | IF(1:500) |
| Antibody | anti-Rabbit IgG, Alexa Fluor 546 (Goat polyclonal) | Thermo Fisher Scientific | A-11035 | IF(1:500) |
| Antibody | anti-Rabbit IgG, Alexa Fluor 647 (Goat polyclonal) | Thermo Fisher Scientific | A-21245 | IF(1:500) |
| Antibody | anti-Rabbit IgG, Atto 594 (Goat polyclonal) | Rockland Immunochemicals Inc. | 611-155-122 | IF(1:500) |
| Antibody | anti-Rabbit IgG, Star Red (Goat polyclonal) | abberior GmbH | STRED-1002–500 UG | IF(1:500) |
| Cell line (*Homo sapiens*) | Human embryonic kidney (HEK) 293T | ATCC | CRL-3216 | |
| Chemical compound, drug | 2,4,6-Triiodophenol (TIP) | Thermo Fisher Scientific | AAA1714506 | 4 µM |
| Chemical compound, drug | 20 x SSC | Thermo Fisher Scientific | AM9763 | |
| Chemical compound, drug | Cycloheximide (CHX) | Sigma-Aldrich | C4859 | 50 nM |
| Chemical compound, drug | DABCO | Sigma-Aldrich | D27802 | |

*Appendix 1 Continued on next page*

*Appendix 1 Continued*

| Reagent type (species) or resource | Designation | Source or reference | Identifiers | Additional information |
|---|---|---|---|---|
| Chemical compound, drug | D-AP5 | Tocris | 0106 | 10 µM |
| Chemical compound, drug | Dimethyl sulfoxide (DMSO), Anhydrous | Thermo Fisher Scientific | D12345 | |
| Chemical compound, drug | Glycine | Tocris | 0219 | 0.2 mM |
| Chemical compound, drug | Jasplakinolide (Jsp) | Tocris | 2792 | |
| Chemical compound, drug | $JF_{549}$-HTL | *Grimm et al., 2015* | N/A | 10 nM |
| Chemical compound, drug | $JF_{549}$i-HTL | *Xie et al., 2017* | N/A | 10 nM |
| Chemical compound, drug | $JF_{646}$-HTL | *Grimm et al., 2015* | N/A | 20 nM |
| Chemical compound, drug | Latrunculin A (LatA) | Tocris | 3973 | 1 µM |
| Chemical compound, drug | Lipofectamine 3000 | Thermo Fisher Scientific | L3000001 | |
| Chemical compound, drug | MNI-caged-L-glutamate (MNI) | Tocris | 14-905-0 | 2 mM |
| Chemical compound, drug | Mowoil 4–88 | Sigma-Aldrich | 813831 | |
| Chemical compound, drug | MyoVin-1 | Sigma-Aldrich | 475984 | 10 µM |
| Chemical compound, drug | NbActiv4 Minus Phenol Red | Transnetyx | NB4PR500 | |
| Chemical compound, drug | Nocodazole | Tocris | 1228 | 500 nM |
| Chemical compound, drug | Normal Goat Serum | Thermo Fisher Scientific | 31873 | |
| Chemical compound, drug | Papain | Worthington Biochemical Corporation | LK003176 | |
| Chemical compound, drug | Paraformaldehyde (PFA) | Electron Microscopy Sciences | 15713 | |
| Chemical compound, drug | Pentabromopseudilin (PBP) | Sigma-Aldrich | SML2428 | 4 µM |
| Chemical compound, drug | Phalloidin Star 635 p | abberior GmbH | ST635P-0100–20 UG | 0.17 µM |
| Chemical compound, drug | Poly-ᴅ-Lysine (PDL) Hydrobromine | Sigma-Aldrich | P6407 | |
| Chemical compound, drug | ProLong Diamond Antifade Mountant with DAPI | Thermo Fisher Scientific | P36971 | |
| Chemical compound, drug | Saponin | Sigma-Aldrich | 47036 | |
| Chemical compound, drug | Tetrodotoxin Citrate (TTX) | Tocris | 1069 | 200 nM |
| Chemical compound, drug | Triton X-100 | Sigma-Aldrich | T8787 | |
| Commercial assay or kit | HCR RNA-FISH Bundle for HaloTag and *gria1* mRNA | Molecular Instruments | N/A | |
| Commercial assay or kit | P3 Primary Cell 96-well Nucleofector Kit (96 RCT) | Lonza | V4SP-3096 | |
| Recombinant DNA reagent | CAG-tdTomato (plasmid) | James Wilson | Addgene #105554 | |
| Recombinant DNA reagent | hSyn-GEM (plasmid) | This paper | N/A | Plasmid for expressing GEM-HT-ST in neurons |

*Appendix 1 Continued on next page*

*Appendix 1 Continued*

| Reagent type (species) or resource | Designation | Source or reference | Identifiers | Additional information |
|---|---|---|---|---|
| Recombinant DNA reagent | hSyn-GFP (plasmid) | James Wilson | Addgene #105539 | |
| Recombinant DNA reagent | hSyn-tractin (plasmid) | This paper | N/A | Plasmid for expressing F-tractin-mNeongreen in neurons |
| Recombinant DNA reagent | ITPKA-mNeongreen (plasmid) | *Chertkova et al., 2017* | Addgene # 98883 | |
| Recombinant DNA reagent | LZ10 PBREBAC-H2BHalo (plasmid) | *Li et al., 2016* | Addgene #91564 | |
| Recombinant DNA reagent | pAS1NB c Rosella I (plasmid) | *Rosado et al., 2008* | Addgene #71245 | |
| Recombinant DNA reagent | pBAD/His-miRFP670 (plasmid) | *Shcherbakova et al., 2016* | Addgene #79884 | |
| Recombinant DNA reagent | pCAGGS (plasmid) | *Niwa et al., 1991* | BCCM/LMBP #2453 | |
| Recombinant DNA reagent | pCAG-GluA1-P2A-GFP (plasmid) | This paper | N/A | Plasmid for expressing GluA1-P2A-GFP in HEK293T cells for patching assays |
| Recombinant DNA reagent | pCAG-GluA1-HT-P2A-GFP (plasmid) | This paper | N/A | Plasmid for expressing GluA1-HT-P2A-GFP in HEK293T cells for patching assays |
| Recombinant DNA reagent | pCAG-GluA1-HT-SEP-P2A-mScarlet (plasmid) | This paper | N/A | Plasmid for expressing GluA1-HT-SEP-P2A-mScarlet in HEK293T cells for patching assays |
| Recombinant DNA reagent | pCMV-PfV-Sapphire-IRES-DSRed (plasmid) | *Delarue et al., 2018* | Addgene #116934 | |
| Recombinant DNA reagent | pCMV2-SEP-GluA1 (plasmid) | *Blanco-Suarez and Hanley, 2014* | Addgene #64942 | |
| Recombinant DNA reagent | pEGFP-C1 (plasmid) | N/A | Clontech | |
| Recombinant DNA reagent | pEGFP-MyosinVa-Ctail (plasmid) | *Correia et al., 2008* | Addgene #110169 | |
| Recombinant DNA reagent | pEGFP-MyosinVb-Ctail (plasmid) | *Correia et al., 2008* | Addgene #110170 | |
| Recombinant DNA reagent | pEGFP-MyosinVI-Ctail (plasmid) | This paper | N/A | Plasmid for expressing the dominant-negative myosin VI C-tail to inhibit myosin VI function |
| Recombinant DNA reagent | PX551 (plasmid) | *Swiech et al., 2015* | Addgene #60957 | |
| Recombinant DNA reagent | PX552 (plasmid) | *Swiech et al., 2015* | Addgene #60958 | |
| Recombinant DNA reagent | px552-sg-gria1-HT (plasmid) | This paper | N/A | The donor plasmid for knocking HaloTag into *Gria1* via HITI |
| Recombinant DNA reagent | px552-sg-gria1-HT-SEP (plasmid) | This paper | N/A | The donor plasmid for knocking HaloTag-SEP into *Gria1* via HITI |
| Recombinant DNA reagent | Cas9 target sequence for sgRNA insert | Integrated DNA Technologies | N/A | 5'-ACCCTCCCGAG ACCATACCAGGG-3' 5'-AACCCCTGGTATG GTCTCGGGAG-3' |
| Recombinant DNA reagent | Sequencing primers for 5' insertion site of HaloTag into *Gria1* | Integrated DNA Technologies | N/A | 5'-ATGCCTGCAGGTC GAAGAGACCTACCCA GACGATG-3' 5'-GTGAATTCGAGCTCG GGCGGATGAACTCCATAAATGC-3' |
| Recombinant DNA reagent | Sequencing primers for 3' insertion site of HaloTag into *Gria1* | Integrated DNA Technologies | N/A | 5'-ATGCCTGCAGGTCGA TGCATTTATGGAGTTCATCCGC-3' 5'-GTGAATTCGAGCTCGT GTGTTCTGCTGTTCAGGAG-3' |
| Software, algorithm | Directional Bias | This paper | N/A | MATLAB script used to determine the directional bias of particles |
| Software, algorithm | Fiji | *Schindelin et al., 2012* | N/A | |
| Software, algorithm | FISH-Quant | *Mueller et al., 2013* | https://fish-quant.github.io/ | |

*Appendix 1 Continued on next page*

*Appendix 1 Continued*

| Reagent type (species) or resource | Designation | Source or reference | Identifiers | Additional information |
|---|---|---|---|---|
| Software, algorithm | HMM-Bayes, v1 | *Monnier et al., 2015*; *Bathe BioNanoLab, 2021* | https://github.com/lcbb/HMM-Bayes | MATLAB script used to predict motion states of particles |
| Software, algorithm | Imaris 9.3 | Bitplane (Belfast, United Kingdom) | N/A | |
| Software, algorithm | Impsector | abberior GmbH (Göttingen, Germany) | N/A | |
| Software, algorithm | MATLAB 2016a | Mathworks (Natick, MA) | https://www.mathworks.com/ | |
| Software, algorithm | MSDanalyzer, v1.3 | *Tarantino et al., 2014*; *Tinevez, 2020* | https://github.com/tinevez/msdanalyzer | MATLAB script used to determine diffusion coefficients of particles |
| Software, algorithm | Prism 9, v8 | GraphPad (San Diego, CA) | https://www.graphpad.com/ | |
| Software, algorithm | SLIMfast | This paper | N/A | MATLAB program for 2D single-particle tracking |
| Software, algorithm | Zen Black | Zeiss (Jena, Germany) | N/A | |
| Strain, strain background (*AAV*) | AAV2/9-CAG-TdTomato | Janelia Research Campus Viral Services | N/A | |
| Strain, strain background (*AAV*) | AAV2/9-hSyn-GCaMP6s | Janelia Research Campus Viral Services | N/A | |
| Strain, strain background (*AAV*) | AAV2/9-hSyn-GEM | Janelia Research Campus Viral Services | N/A | |
| Strain, strain background (*AAV*) | AAV2/9-hSyn-GFP | Janelia Research Campus Viral Services | N/A | |
| Strain, strain background (*AAV*) | AAV2/9-hSyn-tractin | Janelia Research Campus Viral Services | N/A | |
| Strain, strain background (*AAV*) | rAAV2-PX551 | Janelia Research Campus Viral Services | N/A | |
| Strain, strain background (*AAV*) | rAAV2-px552-sg-gria1-HT | Janelia Research Campus Viral Services | N/A | |
| Strain, strain background (*AAV*) | rAAV2-px552-sg-gria1-HT-SEP | Janelia Research Campus Viral Services | N/A | |
| Strain, strain background (*AAV*) | rh10-PX551 | Janelia Research Campus Viral Services | N/A | |
| Strain, strain background (*AAV*) | rh10-px552-sg-gria1-HT | Janelia Research Campus Viral Services | N/A | |
| Strain, strain background (*AAV*) | rh10-px552-sg-gria1-HT-SEP | Janelia Research Campus Viral Services | N/A | |
| Strain, strain background (*Rattus norvegicus*) | Sprague-Dawley rats, CD1 (Sprague-Dawley postnatal pups P0, M and F) | Charles River Laboratories | Strain code: 400 | |
| Strain, strain background (*Escherichia coli*) | Stbl3 | Invitrogen | C737303 | |

