## [Editor Report]

In this manuscript, the authors developed a sensitive single particle tracking method for endogenous AMPA receptors. They found that AMPAR-containing vesicles showed reduced mobility near stimulation sites, due to increased F-actin bundling in dendritic shafts. The study provides compelling evidence on a new important mechanism of AMPAR trafficking using state-of-the-art labeling and analysis techniques.

---

## [Decision Letter]

**Decision letter after peer review:**

Thank you for submitting your article "Plasticity-induced actin polymerization in the dendritic shaft regulates intracellular AMPA receptor trafficking" for consideration by *eLife*. Your article has been reviewed by 3 peer reviewers, and the evaluation has been overseen by a Reviewing Editor and Lu Chen as the Senior Editor. The reviewers have opted to remain anonymous.

Essential revisions (for the authors):

1) The new technology needs to be better validated.

1a: The rate of point mutations and indel formation by HITI-knockin are not validated. Currently, the authors validated the gene insertion only by in situ hybridization. This caveat should be discussed.

1b: Colocalization assay of HT-GluA1 and GluA1 should be performed with better quality and other receptors.

1c: The activity of GluA1-HT-SEP and GluA1-HT need to be characterized using heterologous cells lacking endogenous AMPARs.

2) Uncaging experiments (Figure 3) appear to be performed in poor culture conditions. Also, the stimulation condition is not well calibrated.

2a: The representative figure shows non-spiny, non-healthy neurons. It is not clear whether all experiments are performed under such poor conditions. The reviewers want to see all raw images of uncaging experiments.

2b: Experiments should be performed on spiny, healthy neurons, as neuronal morphology (or a cell type) can strongly influence the experimental outcome.

2c: The uncaging strength appeared to be not characterized. It is required to see if stimulation is spine-specific using ca^2+^ imaging or electrophysiology.

3) Experiments using Tractin (Figure 4) needs to be validated with a different method, as tractin can cause an artifact.

4) Additional validation (Figure 5), such as shRNA or CRISPR-knockdown, is required for the involvement of myosin (currently only a single pharmacological condition is used).

5) The relevance of the finding with LTP should be toned down. (see reviewer 3)

*Reviewer #1 (Recommendations for the authors):*

This story relies on the fact that Halo tag protein faithfully represents the GluA1 protein localization. However, Figure 1E (immunostaining) only represents a correlation between these two signals. There seem to be many green and red signal where they don't co-localize, and even in the yellow part, it is hard with this resolution to know how much they co-localize. To address this issue authors can use better microscopy such as the airy scan system that they used for F-actin imaging, and also show that GluA1 and halo co-localize significantly more than other synaptic receptors (e.g. halo and NMDAR).

This story also relies on the fact that the author's system can clearly distinguish GluA1-HT vesicles from surface GluA1-HT (line 208). Authors claim that this separation was based on 1) its localization on the shaft, 2) bleaching characteristics of puncta signal, and 3) its motion type. For 1), as the authors themselves show in Figure 6, surface GluA1-HT (GluA1-HT-SEP) can exist in the shaft so this is not a good indicator. Also synapses can be often been in the what seems to be in the shaft area with poor optical sectioning. For 2), Figure 2- supplemental 1 does not describe the distinct differences between bleaching behavior of GluA1-HT vesicles from surface GluA1-HT. Did authors perform this type of bleaching experiment for all signals, categorize them, and then move on to tracking? If so, how? As is now, it is useless, it can be left out. It is indeed hard to imagine that the fluorescence declines in a stepwise fashion in a timely fashion.

versus DV is the most critical concept of this paper analyzed in the rest of the study. The authors argue that they correspond to diffusion versus active transport but they did not provide any evidence for this and it is not clear at all how they concluded in this way. It is also not clear how motion type was used to distinguish the two populations. They could for example use, nocodazole or latrunculin to see if the DV population disappears. For this parameter to be used, there must be little overlap between the motion of GluA1-HT vesicles and surface GluA1-HT. For example, in Figure 7 supplements C and D, how did the authors distinguish diffusion versus active? Authors could use surface GluA1-HT (using a membrane-impermeable tag or SEP) motion to analyze the data. The above point is very important as surface GluA1 is also known to be trapped upon LTP (Opazo et al. 2010) and we need to be sure that authors are looking at GluA1-HT vesicles.

The neurons used in some of the experiments are in poor condition. In Figure 3A, the bright spot where uncaging was performed does not look like a spine. It does not make sense that the spine would look so much brighter than the dendrite when the cell is just labeled with cytosolic GFP. Did authors really uncage at dendritic spines? I would like to see all other original images of the uncaging experiment so that I am convinced. Also, the authors need to establish a proper uncaging condition where the stimulation of a single spine elicits epsc of reasonable size in the cell body. How long was the spine enlargement maintained? In order to say LTP, the change must be maintained at least for 30 min or so but I cannot see the information. When was the image taken after uncaging? It may be hidden somewhere in the method but the authors need to make an effort to make it more clear.

*Reviewer #2 (Recommendations for the authors):*

1. Introduction, lane 26: There has been a lot of recent work on molecular mechanisms of LTP. Citing 2013 paper to say that there is not much known is not up to date.

2. Figure 1D: to convincingly demonstrate the colocalization please include zoom-ins or higher magnification images.

3. Figure 2E (also Figure 4): where have entries of GluA1-vesicles into dendritic spines been observed or were they removed from the analysis? I also have not found them in other given examples and videos.

4. Motility analysis: I have no critical comments here, nevertheless I would like to acknowledge that the authors did a great job, and I am sure there will be many labs who would benefit from their analysis pipeline.

5. Lane 240: term sub-diffusion is somewhat confusing. Perhaps replace by confined diffusion?

6. Figure 3A, 3C: images are not convincing. The dendrite where uncaging has been done looks aspiny and neighbouring branches are full of GFP aggregation and have varicosities. If this is a representative example as the figure legend says, I have some concerns about the quality of the experiment.

7. Figure 3, legend: what does n=9 mean? 9 spines, or 9 cells? How many independent cultures? This experiment has to be improved and redone.

8. Figure 3B: for post-LTP time should be indicated. Please include a time-lapse to show that there is a persistent change in the spine volume upon uncaging.

9. Figure 3C: What is the rationale behind defining dendritic zones? Is it based on pre-existing knowledge from the literature, ie. the length of dendrite within which synapses can cluster? Also, in Figure 2 Homer-GFP is used as a synaptic marker. What is the reason to switch to the GFP cell fill in Figure 3?

10. Figure 3: Glutamate uncaging results in elevated calcium levels. Depending on the stimulation strength it is limited to the activated spine or spreads out into a parent dendrite. I am wondering how far calcium signals spread in these experiments. This would be important to know as actin dynamics and myosin activity is controlled by calcium signalling. Is there a correlation between calcium spread with an area of increased actin polymerization and a zone with reduced AMPAR vesicle motility?

11. Figure 4, tractin experiments: I am not very convinced by the use of tractin as an F-actin probe. In the given representative images, there are very obvious actin cables in dendrites upon overexpression of F-tractin. This might be an artifact induced by overexpression. Please include another method of actin labeling. There are very nice recent tools available (i.e. CRISPR-based labeling of endogenous actin). Although they also label G-actin, when imaged at high magnification various F-actin-based structures are quite easy to visualize.

12. Figure 4A: as mentioned above, these are more like actin cables and bundles seen in dendrites. This is not typical. Can such redistribution be reproduced by staining endogenous F-actin? For example, with fluorescently labelled Phalloidin?

13. Figure 5A: please include example traces.

14. Figure 6C: please include examples.

15. To stringent the myosin part, in addition to inhibitors, myosin shRNAs or dominant negative constructs could be used.

16. Figure legends: please specify what n means in each case (number of cultures, cells, synapses…).

*Reviewer #3 (Recommendations for the authors):*

Line 4. This is the first manuscript to show that GluA1 moves near activated synapses intracellularly, as the authors acknowledged on lines 54-56. Thus, it is more appropriate to describe "but whether and how AMPAR-containing vesicles……" on line 4-a similar comment for Line 44.

Line 77. Some references seem incorrect. Please double-check.

One example is Hardin et al. (Line 77) because this paper does not describe PSD. A paper (PMID: 11082065) might fit better.

Line 165. "79% of cells express Gria1". Does this mean that 21% of cells express truncated GluA1 due to CRISPR or are knocked out?

Line 195. The authors conclude no significant alteration in GluA1-HT function based on no significant difference in Figure 1G. However, with this comprehensive standard error, it is challenging to conclude no biological difference. The authors need to show this with a more robust system, for example, transfected HEK cells that do not express AMPARs endogenously.

Line 202. Figure 2A. It is helpful to understand the significance of this method by showing the JF646 channel for comparison of steady vs. block-chase protocol.

Line 242. I agree that neuronal activity induces GluA1 trafficking to synapses, but not sure whether the phenomenon ties with LTP. During LTP, only AMPAR but not NMDAR is increased. If the authors aim to tie the phenomena to LTP, the authors need to show AMPAR specificity. Otherwise, tone down.

Line 316. Overall, the discovery of F-actin involvement is interesting, and the result is robust. However, if the authors try to tie this to AMPAR insertion during LTP, it is critical to show the dynamic of NMDAR or other receptors that do not move upon neuronal activation.

Line 409. As the authors conclude, "actin polymerization as the mechanism that mediates stimulation-dependent GluA1 vesicle confinement" that is observed upon the neuronal activity, testing effects of actin polymerization as well as myosin activation on excitatory synaptic transmission broaden the impact of this study, if the authors meant to propose this as one of LTP mechanisms.

Line 467. As stated in the comment on line 195, the authors need to show the activity of GluA1-HT-SEP using heterologous cells lacking endogenous AMPARs.

Figures. Images for colocalization require high magnification of one dendrite as an inset. For example, I thought that GluA1 and GluA1-HT should colocalize perfectly in Figure 1E, but it looks not.

Discussion can be shortened to highlight the main points. Since this is the first manuscript to show a novel robust approach, it is better to discuss both pros and cons of this approach.

---

## [Author Response]

Essential revisions (for the authors):1) The new technology needs to be better validated.1a: The rate of point mutations and indel formation by HITI-knockin are not validated. Currently, the authors validated the gene insertion only by in situ hybridization. This caveat should be discussed.

To determine the rates of indel formation and point mutations introduced by HITI-mediated gene editing, we sequenced the 5’ and 3’ ends of HaloTag (into the genomic copy of Gria1) in knock-in cells. We find that indels occur at a low rate (9.8% of HaloTag knock-in cells contain an indel at the 5’ end and 11.2% at the 3’ end) and that HaloTag is inserted in the correct orientation. Likewise, point mutations occurred with low frequency (2.4% of HaloTag knock-in cells contained a point mutation at the 5’ end and 2.8% at the 3’ end). We have included this information, as well as the technique used to sequence HaloTag insertions, in Figure 1—figure supplement 1D and in the methods (Methods, lines 2198-2212). In addition, we further address the drawbacks of using HITI in the discussion (Discussion, lines 571-578).

1b: Colocalization assay of HT-GluA1 and GluA1 should be performed with better quality and other receptors.

We have repeated co-immunofluorescence labeling experiments for HaloTag (GluA1-HT) and GluA1, and imaged samples using stimulation emission depletion (STED) microscopy with a pixel size of 20 nm (we were able to distinguish structures with a resolution of 60-75 nm depending on the emission wavelength). We have included high resolution images in Figure 1—figure supplement 3A. We demonstrate strong colocalization between HaloTag and GluA1 in edited neurons at these length scales. Moreover, we performed co-immunofluorescence labeling assays on GluA1-HT/untagged GluA1 and other postsynaptic density markers, including NMDARs (Figure 1—figure supplement 3B-D). The colocalization between HaloTag and GluA1 is much stronger than between HaloTag and NMDARs. We find that both HaloTag and GluA1 exhibit poor colocalization to NMDARs at length scales resolvable by STED, similar to what others have described previously (Goncalves et al., 2020).

1c: The activity of GluA1-HT-SEP and GluA1-HT need to be characterized using heterologous cells lacking endogenous AMPARs.

We have characterized the activity of GluA1-HT and GluA1-HT-SEP by expressing these receptors in HEK293T cells, which do not express GluA1, and performing whole-cell patch clamping assays (Figure 1—figure supplement 4 and Figure 5—figure supplement 2). We find that both GluA1-HT and GluA1-HT-SEP respond to glutamate application in a similar manner to untagged GluA1 in HEK293T cells.

2) Uncaging experiments (Figure 3) appear to be performed in poor culture conditions. Also, the stimulation condition is not well calibrated.2a: The representative figure shows non-spiny, non-healthy neurons. It is not clear whether all experiments are performed under such poor conditions. The reviewers want to see all raw images of uncaging experiments.

We thank the reviewers for pointing out potential issues with the health of neurons during uncaging experiments. We have included all raw images for uncaging experiments.

2b: Experiments should be performed on spiny, healthy neurons, as neuronal morphology (or a cell type) can strongly influence the experimental outcome.

We have replaced spine images in Figure 3 (now Figure 2) with images from a spiny neuron. We have also removed spine expansion and single particle tracking (SPT) data from neurons that appear unhealthy, and performed reanalysis using new data from neurons that have healthy morphologies (raw images included). These results are consistent with our previous observations.

2c: The uncaging strength appeared to be not characterized. It is required to see if stimulation is spine-specific using ca^2+^ imaging or electrophysiology.

We have now included a supplemental figure describing the calibration of MNI-glutamate uncaging using GCaMP6s (a fluorescence-based calcium sensor; Figure 2—figure supplement 1). We demonstrate that the laser strength we used for MNI-glutamate uncaging specifically induces calcium influx at the targeted spine. Moreover, we show that spine expansion is induced specifically at the targeted spine by our uncaging protocol, and not at neighboring spines. We have added additional detail to describe our photostimulation protocols in the methods (Methods, lines 2420-2449).

3) Experiments using Tractin (Figure 4) needs to be validated with a different method, as tractin can cause an artifact.

To address this issue, we have performed phalloidin labeling on fixed neurons after cLTP stimulation and imaged phalloidin with STED microscopy (Figure 3—figure supplement 2C). We find that neurons treated with cLTP have greater phalloidin labeling, and also exhibit more F-actin fibers. Neurons that are treated with LatA during cLTP have very poor labeling, demonstrating that cLTP stimulates actin polymerization in the dendritic shaft.

4) Additional validation (Figure 5), such as shRNA or CRISPR-knockdown, is required for the involvement of myosin (currently only a single pharmacological condition is used).

Portions of Figure 5 describing the effect of myosin inhibition on vesicle motion have been modified and moved to the supplemental materials (now Figure 3—figure supplement 6) to improve the clarity of the main figures and text. To better validate the role of myosin, we have used dominant negative C-terminal domains of myosin Va, Vb, and VI (Correia et al., 2008) to test whether inhibiting cargo interactions between GluA1-HT vesicles and myosin disrupt cLTP-mediated reductions in motion (Figure 3—figure supplement 6). Similar to pharmacological myosin inhibition, we find that disrupting myosin did not attenuate the reduction of GluA1-HT vesicle motion in response to cLTP. In addition, we recapitulate the effect of pharmacological myosin inhibition on the directional bias of GluA1-HT vesicles by showing that expressing the dominant negative domains of myosin Va and Vb (but not VI) disrupts the change in directional bias we observe after cLTP (Figure 6—figure supplement 2G).

5) The relevance of the finding with LTP should be toned down. (see reviewer 3)

We have toned down the relevance of our findings to LTP throughout the text, and instead modified the text to focus on our results in the context of synaptic plasticity. More specifically, we clarify that our protocols stimulate neuronal activity, and that we induce structural plasticity by MNI-glutamate uncaging. Moreover, we clarify the difference between structural and functional plasticity (Introduction, lines 51-54, and Results, lines 235-240).

Reviewer #1 (Recommendations for the authors):- This story relies on the fact that Halo tag protein faithfully represents the GluA1 protein localization. However, Figure 1E (immunostaining) only represents a correlation between these two signals. There seem to be many green and red signal where they don't co-localize, and even in the yellow part, it is hard with this resolution to know how much they co-localize. To address this issue authors can use better microscopy such as the airy scan system that they used for F-actin imaging, and also show that GluA1 and halo co-localize significantly more than other synaptic receptors (e.g. halo and NMDAR).

We have repeated co-immunofluorescence labeling experiments for HaloTag and GluA1, and imaged with STED microscopy. By applying gaussian fits to fluorescence intensity line profiles drawn through GluA1 and HaloTag signals, we resolved spots separated by less than 100 nm, which is less than the average distance that we and others have observe between GluA1 and other postsynaptic density markers (Figure 1—figure supplement 3A; Goncalves et al., 2020). At these length scales, we find very strong colocalization between HaloTag and GluA1 in knock-in cells. As an internal control, we find weak colocalization between HaloTag/untagged GluA1 and NMDARs (Figure 1—figure supplement 3B-D).

- This story also relies on the fact that the author's system can clearly distinguish GluA1-HT vesicles from surface GluA1-HT (line 208). Authors claim that this separation was based on 1) its localization on the shaft, 2) bleaching characteristics of puncta signal, and 3) its motion type. For 1), as the authors themselves show in Figure 6, surface GluA1-HT (GluA1-HT-SEP) can exist in the shaft so this is not a good indicator. Also synapses can be often been in the what seems to be in the shaft area with poor optical sectioning. For 2), Figure 2- supplemental 1 does not describe the distinct differences between bleaching behavior of GluA1-HT vesicles from surface GluA1-HT. Did authors perform this type of bleaching experiment for all signals, categorize them, and then move on to tracking? If so, how? As is now, it is useless, it can be left out. It is indeed hard to imagine that the fluorescence declines in a stepwise fashion in a timely fashion.

We used highly laminated optical lightsheet (HILO) microscopy to image volumes surrounding the center of the dendritic shaft (Methods, lines 2342-2347). In HILO microscopy, the laser is applied to sample at a sharp angle to improve optical sectioning and suppress out of focus signals from outside the illuminated volume (Tokunaga et al., 2008). However, we agree it is difficult to fully exclude signal from the surface, especially because the dendrite is not always flat. We have performed additional experiments to better distinguish surface GluA1-HT from vesicular GluA1-HT (Figure 1—figure supplement 6C). First, we tracked and characterized GluA1-HT labeled with cell-membrane impermeable JF_549_i-HTL (i.e., only surface GluA1-HT). We find that these particles bleach in 4 steps or fewer, within 100 frames (Figure 1—figure supplement 6B, images and line graph). Importantly, almost none of these particles exhibit active transport, demonstrating that they are not in vesicles (Figure 1—figure supplement 6C, bar graph), and generally have a higher diffusion coefficient than all GluA1-HT particles (0.22 ± 0.23 μm^2^/s versus 0.16 ± 0.17 μm^2^/s; Figure 1—figure supplement 6C, histogram). In addition, to exclude signal that might be clusters of GluA1-HT trapped in synapses (but appear in the dendritic shaft because of poor optical sectioning), we have tracked GluA1-HT particles that colocalize to GFP-Homer1c in neurons that contain GluA1-HT knock-ins and also express GFP-Homer1c. We find GluA1-HT particles that colocalize with GFP-Homer1c have significantly lower diffusion coefficients than all GluA1-HT particles (0.01 ± 0.009 μm^2^/s; Figure 1—figure supplement 7). Consequently, we filtered out all particles with a diffusion coefficient below 0.02 μm^2^/s from our analysis. We have added greater details for how vesicles were identified into the methods (Methods, lines 2373-2389). Using our particle tracking pipeline (MatLab code available in the materials), we first manually inspected all spots to ensure tracks are accurately reconstructed. We then removed particles clearly localized inside spines for the majority of steps in their trajectories. We then filtered all particles that exhibit active transport based on HMM-Bayes analysis. After this, we examined all particles that exhibit only diffusive motion, and further filtered out particles that bleach in fewer than 100 frames. Lastly, we filtered particles based on their diffusion coefficient. We considered particles with a diffusion coefficient greater than 0.45 μm^2^/s (one standard deviation greater than the average diffusion coefficient for GluA1-HT on the cell surface) to be on the cell surface and below 0.02 μm^2^/s to possibly be receptors inside PSDs.

- D versus DV is the most critical concept of this paper analyzed in the rest of the study. The authors argue that they correspond to diffusion versus active transport but they did not provide any evidence for this and it is not clear at all how they concluded in this way. It is also not clear how motion type was used to distinguish the two populations. They could for example use, nocodazole or latrunculin to see if the DV population disappears. For this parameter to be used, there must be little overlap between the motion of GluA1-HT vesicles and surface GluA1-HT. For example, in Figure 7 supplements C and D, how did the authors distinguish diffusion versus active? Authors could use surface GluA1-HT (using a membrane-impermeable tag or SEP) motion to analyze the data. The above point is very important as surface GluA1 is also known to be trapped upon LTP (Opazo et al. 2010) and we need to be sure that authors are looking at GluA1-HT vesicles.

We used HMM-Bayes to infer whether a particle exhibits diffusion or active transport based on the step size and processivity of the particle (Monnier, 2015). HMM-Bayes was previously validated by applying it to infer the motion of simulated particles and particles with characterized motion (Monnier, 2015). We also tested the accuracy of HMM-Bayes by comparing parameters calculated by HMM-Bayes versus MSDanalyzer (Tarantino et al., 2014; Figure 1—figure supplement 5E).

Nevertheless, we agree that further experimental evidence is important to demonstrate inferences about motion states made by HMM-Bayes are accurate. To address this issue, we have tracked GluA1-HT particles labeled with cell-membrane impermeable JF_549_i-HTL as they should not exhibit active transport (i.e., GluA1-HT particles on the cell surface; Figure 1—figure supplement 6C). In these experiments, very few particles exhibit DV and on average have a higher diffusion coefficient than particles labeled with cell-membrane permeable JF_549_-HTL (i.e., particles both on the surface and in vesicles and the secretory pathway). GluA1-HT-JF_549_i particles that exhibit DV likely reflect those that have been endocytosed. Particles that colocalize with GFP-Homer1c have very low diffusion coefficients and thus were excluded from analysis (Figure 1—figure supplement 7, see previous comment). To further demonstrate that particles exhibiting DV require motor-based transport, we treated neurons with Nocodazole, and find a near total loss of particles exhibiting DV parallel to the length of the dendrite (Figure 1—figure supplement 5H). Although we demonstrate that LatA treatment blocks short-ranged active transport of vesicles to their exocytic sites (Figure 6 and Figure 6—figure supplement 2), it does not disrupt active transport along the length of dendritic shaft and thus would not be an appropriate treatment to demonstrate that particles exhibiting DV require active transport.

In the case of Figure 7—figure supplement 1 (now called Figure 6—figure supplement 1A), we find HMM-Bayes sometimes predicts a multi-state particle exhibits diffusion despite having highly direct motion, albeit with relatively low probability. This is exclusively an issue with multi-state vesicles that appear to have directed motion but with small step sizes. In other words, HMM-Bayes can sometimes underestimate the processivity of multi-state vesicles, and assign diffusion rather than active transport. To address this issue, we segmented multi-state vesicles that exhibit ambiguity in their motion and analyzed each state separately using HMM-Bayes and MSDanalyzer (Tarantino et al., 2014; Methods, lines 2366-2370). We have reassigned these multi-state vesicles based on whether they exhibit DV. To avoid confusion, we have changed the figure to reflect multi-state vesicles that exhibit either purely diffusive motion or a combination of diffusion and active transport rather than classify vesicles based on their diffusion coefficient (Figure 6—figure supplement 1A). Lastly, we now abbreviate active transport as “AT” to avoid conflating DV with active transport.

- The neurons used in some of the experiments are in poor condition. In Figure 3A, the bright spot where uncaging was performed does not look like a spine. It does not make sense that the spine would look so much brighter than the dendrite when the cell is just labeled with cytosolic GFP. Did authors really uncage at dendritic spines? I would like to see all other original images of the uncaging experiment so that I am convinced. Also, the authors need to establish a proper uncaging condition where the stimulation of a single spine elicits epsc of reasonable size in the cell body. How long was the spine enlargement maintained? In order to say LTP, the change must be maintained at least for 30 min or so but I cannot see the information. When was the image taken after uncaging? It may be hidden somewhere in the method but the authors need to make an effort to make it more clear.

We have removed images of unhealthy dendrites from Figure 3 (now Figure 2) and replaced them with images from a healthy, spiny dendrite. We have also removed SPT and spine expansion data from any neurons that appear unhealthy, and replaced these with data from healthy neurons. Raw images for uncaging experiments have been included for inspection.

We have toned down the relevance between our results and LTP, and instead focused on structural plasticity. Data on the duration of spine enlargement can now be found in Figure 2—figure supplement 3A. Spine images were taken immediately before and after sLTP stimulation. After taking an image of the post-sLTP spine, we immediately commenced particle imaging. To demonstrate that spine enlargement was persistent, we performed a series of experiments where we imaged the spine and particles 10 minutes after the cessation of sLTP (Figure 2—figure supplement 3A). We have made this information more prominent in the text (Results, lines 244-246 and lines 257-260), and added greater detail describing our photostimulation methods (Methods, lines 2420-2449).

Reviewer #2 (Recommendations for the authors):1. Introduction, lane 26: There has been a lot of recent work on molecular mechanisms of LTP. Citing 2013 paper to say that there is not much known is not up to date.

The reviewer is correct to point out that there have been many recent advances in understanding the molecular mechanism of LTP. However, we have decided to tone down the relevance of our work to LTP and focus on neuronal activity and structural plasticity, and thus this line has been removed. We have added more recent references describing AMPAR trafficking in the context of plasticity, including more recent reviews (Groc and Choquet, 2020; Choquet and Opazo, 2022; Bonnet et al., 2023; Getz et al., 2022).

2. Figure 1D: to convincingly demonstrate the colocalization please include zoom-ins or higher magnification images

We have repeated co-immunofluorescence experiments from Figure 1 (now Figure 1—figure supplement 3) using STED microscopy and we confirm strong colocalization between GluA1 and HaloTag with a resolution of 60-75 nm and included magnified images of the spines.

3. Figure 2E (also Figure 4): where have entries of GluA1-vesicles into dendritic spines been observed or were they removed from the analysis? I also have not found them in other given examples and videos.

We have largely removed signal in spines from analysis, as these signals are immobile and likely represent GluA1-HT particles (not vesicles) that have been captured in postsynaptic densities. We have added supplemental materials to explain this further (Figure 1—figure supplement 7). One exception to this filter is if a vesicle moves from the dendritic shaft into a spine by active transport and if the majority of steps in the trajectory are outside of the spine. Nevertheless, we find very few vesicles are transported directly into spines in our image acquisition. We believe that these events are rare (Tao-Cheng et al., 2011), and that the majority exocytic events occur in extrasynaptic space (Lin et al., 2007; Makino et al., 2009; Patterson et al., 2010). Nevertheless, we cannot rule out the possibility that these events do occur, and indeed because we sparsely label GluA1-HT, we only sample a fraction of the total GluA1-HT vesicles. It would be interesting to further examine the rates of spine entry by GluA1 vesicles in the future.

4. Motility analysis: I have no critical comments here, nevertheless I would like to acknowledge that the authors did a great job, and I am sure there will be many labs who would benefit from their analysis pipeline.

We thank the reviewer for acknowledging our analysis strategy.

5. Lane 240: term sub-diffusion is somewhat confusing. Perhaps replace by confined diffusion?

We have decided to keep the term “subdiffusion” as it has been historically used to describe the plateauing mean square diffusion characteristic of particles with constrained motion (Figure 1—figure supplement 10). Although “confined diffusion” is also an accurate description, we wish to avoid confusing this terminology with “confinement”, which we define in our work as “the restriction of vesicle motion away from its initial position” (Results, lines 211-212). We have defined subdiffusion more clearly in the text (Results, lines 207-209) and included references with its usage (Feder et al., 1996; Saxton, 1999).

6. Figure 3A, 3C: images are not convincing. The dendrite where uncaging has been done looks aspiny and neighbouring branches are full of GFP aggregation and have varicosities. If this is a representative example as the figure legend says, I have some concerns about the quality of the experiment.

We have remade Figure 3 (now Figure 2) to streamline the text and figures. In addition, we have replaced existing images with images of a healthy, spiny dendrite. We have also removed from our analysis any images where the dendrite appears unhealthy or has GFP aggregation. We have included raw images of uncaging experiments for examination.

7. Figure 3, legend: what does n=9 mean? 9 spines, or 9 cells? How many independent cultures? This experiment has to be improved and redone.

We have added more detail to replicates throughout. We have also included more details about replicates in the methods (Methods, lines 2546-2566). We have removed analysis from dendrites that appear unhealthy and replaced with experiments performed on healthy dendrites.

8. Figure 3B: for post-LTP time should be indicated. Please include a time-lapse to show that there is a persistent change in the spine volume upon uncaging.

We have indicated that vesicles were tracked immediately after the cessation for LTP. We have included a supplemental figure to show that changes to spine volume are persistent even 10 minutes after the cessation of uncaging (Figure 2—figure supplement 3).

9. Figure 3C: What is the rationale behind defining dendritic zones? Is it based on pre-existing knowledge from the literature, ie. the length of dendrite within which synapses can cluster? Also, in Figure 2 Homer-GFP is used as a synaptic marker. What is the reason to switch to the GFP cell fill in Figure 3?

We divided the dendrites into 3 equal sections based on proximity to the stimulated spine because we felt this approach would not bias analysis in a particular region of the dendrite (Results, lines 247-250, and Methods, lines 2384-2389). We find that our uncaging protocol does not induce calcium influx or plasticity in neighboring spines (Figure 2—figure supplement 1), and thus felt activity in neighboring spines would not contribute significantly to changing motion in the dendritic shaft near the uncaging site. Consequently, we did not consider the distribution of neighboring spines when segmenting the dendrite. Nevertheless, it would be interesting to further investigate how trafficking in the dendritic shaft is affected based on how synapses are clustered.

We used GFP as a neuronal fill instead of GFP-Homer1c because changes in spine morphology are not readily observed in response to stimulation in cells expressing GFP-Homer1c. This is because spine GFP-Homer1c is stably anchored in postsynaptic densities. Consequently, even as the spine expands, we do not see a dramatic change in the shape of GFP-Homer1c signal. Others have also found that scaffolding protein concentrations do not increase significantly in response to stimulation (Bosch et al., 2014). By contrast, GFP diffuses throughout the spine as it expands, allowing us to easily observe changes in spine size.

10. Figure 3: Glutamate uncaging results in elevated calcium levels. Depending on the stimulation strength it is limited to the activated spine or spreads out into a parent dendrite. I am wondering how far calcium signals spread in these experiments. This would be important to know as actin dynamics and myosin activity is controlled by calcium signalling. Is there a correlation between calcium spread with an area of increased actin polymerization and a zone with reduced AMPAR vesicle motility?

To address this question, we have added a supplemental figure to describe the calibration of MNI-glutamate uncaging (Figure 2—figure supplement 1). Using GCaMP6s, we find that calcium influx is primarily limited to the targeted spine, and thus do not find a correlation between actin polymerization and the spread of calcium signaling. Nevertheless, we cannot rule out the possibility that some calcium below the detection limit of GCaMP6s spreads through the dendritic shaft. Moreover, it is possible that calcium signaling is propagated through the dendritic shaft by CaMKII or other calcium-dependent signaling proteins. It would be interesting to examine the relationship between calcium signal propagation and the origin of actin polymerization in the dendritic shaft.

11. Figure 4, tractin experiments: I am not very convinced by the use of tractin as an F-actin probe. In the given representative images, there are very obvious actin cables in dendrites upon overexpression of F-tractin. This might be an artifact induced by overexpression. Please include another method of actin labeling. There are very nice recent tools available (i.e. CRISPR-based labeling of endogenous actin). Although they also label G-actin, when imaged at high magnification various F-actin-based structures are quite easy to visualize.

We thank the reviewer for bringing this very important point to our attention. See response to Reviewer #2, comment 12 below.

12. Figure 4A: as mentioned above, these are more like actin cables and bundles seen in dendrites. This is not typical. Can such redistribution be reproduced by staining endogenous F-actin? For example, with fluorescently labelled Phalloidin?

Although, F-actin networks have been previously observed in dendrites (E D’Este et al., 2015; Lavoie-Cardinal et al., 2020), the F-actin networks we observe after cLTP appear somewhat different. These differences may be due to tractin, but also because we use a lower resolution imaging technique (Airyscan in this work versus STED in previous studies). Consequently, we have performed F-actin labeling in fixed neurons after cLTP with phalloidin and imaged with STED microscopy (Figure 3—figure supplement 2C). Although the F-actin fibers in the dendritic shaft are not as pronounced, we do observe an increase in F-actin fibers and overall phalloidin labeling in the dendritic shaft after cLTP. This increase in phalloidin labeling is due to increased F-actin, as it can be blocked with LatA treatment.

13. Figure 5A: please include example traces.

We have moved figures related to myosin inhibition into the supplemental materials to streamline the text. This information, as well as experiments with dominant negative myosin C-terminal domains, are now in Figure 3—figure supplement 6. Example traces are included in Figure 3—figure supplement 6A.

14. Figure 6C: please include examples

We have included examples (see Figure 5-video 3).

15. To stringent the myosin part, in addition to inhibitors, myosin shRNAs or dominant negative constructs could be used.

We have repeated cLTP experiments using dominant negative constructs for myosin Va, Vb, and VI (Figure 3—figure supplement 6). Similar to acute pharmacological inhibition, dominant negative inhibition of these motors does not block cLTP-mediated disruption of active transport. However, inhibition of myosin Va and Vb, but not VI, blocks the change in motion bias we observe in response to cLTP (Figure 6—figure supplement 2G).

16. Figure legends: please specify what n means in each case (number of cultures, cells, synapses…).

We have specified n in every figure legend. We have also included more details about replicates in the methods (Methods, lines 2546-2566).

Reviewer #3 (Recommendations for the authors):Line 4. This is the first manuscript to show that GluA1 moves near activated synapses intracellularly, as the authors acknowledged on lines 54-56. Thus, it is more appropriate to describe "but whether and how AMPAR-containing vesicles……" on line 4-a similar comment for Line 44.

We have included this suggestion at Line 4 (now line 26). We have decided not to change the text at Line 44 (now line 73), as Line 44 refers to the synaptic entry of AMPARs rather than AMPAR vesicles.

Line 77. Some references seem incorrect. Please double-check.One example is Hardin et al. (Line 77) because this paper does not describe PSD. A paper (PMID: 11082065) might fit better.

We thank the reviewer for carefully reading our text and bringing this error to our attention. Tardin et al. (2003) is among early works to demonstrate that receptors have reduced diffusion inside synapses, but indeed does not demonstrate PSD proteins as the mechanism underlying this motion change. Consequently, we have moved this reference to line 107 and included the suggested reference.

Line 165. "79% of cells express Gria1". Does this mean that 21% of cells express truncated GluA1 due to CRISPR or are knocked out?

We find that 79% of all cells express *Gria1* from *both* alleles (as opposed to just 1 allele) based on counting active transcription sites (ATS). It is unclear why some neurons express *Gria1* from only one allele, but it may be in part due to the stochastic nature of transcription, which occurs in bursts (Raj and van Oudenaarden, 2008 for review). It is also possible for HITI to introduce indels (Suzuki et al., 2016). Consequently, we have added supplemental material demonstrating that HITI introduces indels into *Gria1* at low frequencies (Figure 1—figure supplement 1D). We also find that the fraction of neurons with two ATS in edited cultures is similar to untransfected cultures (Figure 1—figure supplement 2B), indicating that HITI does not often knock-out or truncate GluA1. Nevertheless, because unwanted mutations are a possibility with this gene-editing technology, we have added discussion about the disadvantages of HITI (Discussion, lines 571-578).

Line 195. The authors conclude no significant alteration in GluA1-HT function based on no significant difference in Figure 1G. However, with this comprehensive standard error, it is challenging to conclude no biological difference. The authors need to show this with a more robust system, for example, transfected HEK cells that do not express AMPARs endogenously.

We have performed electrophysiology experiments in HEK293T cells to show that GluA1-HT respond to glutamate in a similar manner as untagged GluA1 (Figure 1—figure supplement 4).

Line 202. Figure 2A. It is helpful to understand the significance of this method by showing the JF646 channel for comparison of steady vs. block-chase protocol.

In order to streamline the Results section, Figure 2A has been moved to Figure 1—figure supplement 8. We have added the JF_646_ channel to this figure supplement and also to Figure 1-video 1 for comparison.

Line 242. I agree that neuronal activity induces GluA1 trafficking to synapses, but not sure whether the phenomenon ties with LTP. During LTP, only AMPAR but not NMDAR is increased. If the authors aim to tie the phenomena to LTP, the authors need to show AMPAR specificity. Otherwise, tone down.

We have changed the language to reflect GluA1 vesicle trafficking in the context of single spine plasticity rather than long-term potentiation (LTP). We continue to use the cLTP and sLTP as short-hands for the stimulation protocols used to induce plasticity, as these terms have historically been ascribed to these protocols. However, we clarify that cLTP and sLTP are used here to induce neuronal activity and structural plasticity, respectively, rather than LTP. We have included additional references to works which previously demonstrated that these stimulation protocols have been used to probe functional plasticity, and that structural plasticity is strongly associated with functional plasticity (Results, lines 235-240; Matsuzaki et al., 2001; Matsuzaki et al., 2004; Lee et al., 2009; Patterson and Yasuda, 2011; Huganir and Nicoll, 2013; Bosch et al., 2014; Kruijssen and Wierenga, 2019).

Line 316. Overall, the discovery of F-actin involvement is interesting, and the result is robust. However, if the authors try to tie this to AMPAR insertion during LTP, it is critical to show the dynamic of NMDAR or other receptors that do not move upon neuronal activation.

We have changed to text throughout to avoid conflating activity and plasticity with long-term potentiation.

Line 409. As the authors conclude, "actin polymerization as the mechanism that mediates stimulation-dependent GluA1 vesicle confinement" that is observed upon the neuronal activity, testing effects of actin polymerization as well as myosin activation on excitatory synaptic transmission broaden the impact of this study, if the authors meant to propose this as one of LTP mechanisms.

We thank the reviewer for this suggestion. Indeed, finding local actin polymerization in the dendritic shaft as a mechanism underlying LTP would broaden the impact of this study. However, given the technical challenges of demonstrating LTP, we have decided to limit our study to neuronal activity and structural plasticity. Nevertheless, extending our study to LTP would be an important future direction.

Line 467. As stated in the comment on line 195, the authors need to show the activity of GluA1-HT-SEP using heterologous cells lacking endogenous AMPARs.

We have performed electrophysiology experiments in HEK293T cells to show that GluA1-HT-SEP respond to glutamate in a similar manner as untagged GluA1 (Figure 5—figure supplement 2).

Figures. Images for colocalization require high magnification of one dendrite as an inset. For example, I thought that GluA1 and GluA1-HT should colocalize perfectly in Figure 1E, but it looks not.

We have repeated co-immunofluorescence experiments from Figure 1 (now Figure 1—figure supplement 3) using STED microscopy and confirmed strong colocalization between GluA1 and HaloTag at high resolution. Nevertheless, in some cases signals do not colocalize perfectly. For example, if only one *Gria1* allele is edited by HaloTag but both alleles are expressed, there will be some GluA1 signal that does not overlap with HaloTag.

Discussion can be shortened to highlight the main points. Since this is the first manuscript to show a novel robust approach, it is better to discuss both pros and cons of this approach.

We have shortened the discussion overall and also included a paragraph discussing some of the advantages and disadvantage to our labeling strategy (Discussion, lines 561-578).

References

Bonnet, C., Charpentier, J., Retailleau, N., Choquet, D., and Coussen, F. (2023). Regulation of different phases of AMPA receptor intracellular transport by 4.1N and SAP97. *eLife*, *12*, e85609.

Bosch, M., Castro, J., Saneyoshi, T., Matsuno, H., Sur, M., and Hayashi, Y. (2014). Structural and molecular remodeling of dendritic spine substructures during long-term potentiation. *Neuron*, *82*(2), 444–459.

Choquet, D., and Opazo, P. (2022). The role of AMPAR lateral diffusion in memory. *Seminars in cell and developmental biology*, *125*, 76–83.

Correia, S. S., Bassani, S., Brown, T. C., Lisé, M. F., Backos, D. S., El-Husseini, A., Passafaro, M., and Esteban, J. A. (2008). Motor protein-dependent transport of AMPA receptors into spines during long-term potentiation. *Nature neuroscience*, *11*(4), 457–466.

D'Este, E., Kamin, D., Göttfert, F., El-Hady, A., and Hell, S. W. (2015). STED nanoscopy reveals the ubiquity of subcortical cytoskeleton periodicity in living neurons. *Cell reports*, *10*(8), 1246–1251.

Esteves da Silva, M., Adrian, M., Schätzle, P., Lipka, J., Watanabe, T., Cho, S., Futai, K., Wierenga, C. J., Kapitein, L. C., and Hoogenraad, C. C. (2015). Positioning of AMPA Receptor-Containing Endosomes Regulates Synapse Architecture. *Cell reports*, *13*(5), 933–943.

Feder, T. J., Brust-Mascher, I., Slattery, J. P., Baird, B., and Webb, W. W. (1996). Constrained diffusion or immobile fraction on cell surfaces: a new interpretation. *Biophysical journal*, *70*(6), 2767–2773.

Getz, A. M., Ducros, M., Breillat, C., Lampin-Saint-Amaux, A., Daburon, S., François, U., Nowacka, A., Fernández-Monreal, M., Hosy, E., Lanore, F., Zieger, H. L., Sainlos, M., Humeau, Y., and Choquet, D. (2022). High-resolution imaging and manipulation of endogenous AMPA receptor surface mobility during synaptic plasticity and learning. *Science advances*, *8*(30), eabm5298.

Goncalves, J., Bartol, T. M., Camus, C., Levet, F., Menegolla, A. P., Sejnowski, T. J., Sibarita, J. B., Vivaudou, M., Choquet, D., and Hosy, E. (2020). Nanoscale co-organization and coactivation of AMPAR, NMDAR, and mGluR at excitatory synapses. *Proceedings of the National Academy of Sciences of the United States of America*, *117*(25), 14503–14511.

Groc, L., and Choquet, D. (2020). Linking glutamate receptor movements and synapse function. *Science*, *368*(6496), eaay4631.

Hangen, E., Cordelières, F. P., Petersen, J. D., Choquet, D., and Coussen, F. (2018). Neuronal Activity and Intracellular Calcium Levels Regulate Intracellular Transport of Newly Synthesized AMPAR. *Cell reports*, *24*(4), 1001–1012.e3.

Huganir, R. L., and Nicoll, R. A. (2013). AMPARs and synaptic plasticity: the last 25 years. *Neuron*, *80*(3), 704–717.

Kruijssen, D. L. H., and Wierenga, C. J. (2019). Single Synapse LTP: A Matter of Context? *Frontiers in cellular neuroscience*, *13*, 496.

Lavoie-Cardinal, F., Bilodeau, A., Lemieux, M., Gardner, M. A., Wiesner, T., Laramée, G., Gagné, C., and De Koninck, P. (2020). Neuronal activity remodels the F-actin based submembrane lattice in dendrites but not axons of hippocampal neurons. *Scientific reports*, *10*(1), 11960.

Lee, S. J., Escobedo-Lozoya, Y., Szatmari, E. M., and Yasuda, R. (2009). Activation of CaMKII in single dendritic spines during long-term potentiation. *Nature*, *458*(7236), 299–304.

Lin, D. T., Makino, Y., Sharma, K., Hayashi, T., Neve, R., Takamiya, K., and Huganir, R. L. (2009). Regulation of AMPA receptor extrasynaptic insertion by 4.1N, phosphorylation and palmitoylation. *Nature neuroscience*, *12*(7), 879–887.

Makino, H., and Malinow, R. (2009). AMPA receptor incorporation into synapses during LTP: the role of lateral movement and exocytosis. *Neuron*, *64*(3), 381–390.

Matsuzaki, M., Ellis-Davies, G. C., Nemoto, T., Miyashita, Y., Iino, M., and Kasai, H. (2001). Dendritic spine geometry is critical for AMPA receptor expression in hippocampal CA1 pyramidal neurons. *Nature neuroscience*, *4*(11), 1086–1092.

Matsuzaki, M., Honkura, N., Ellis-Davies, G. C., and Kasai, H. (2004). Structural basis of long-term potentiation in single dendritic spines. *Nature*, *429*(6993), 761–766.

Monnier, N., Barry, Z., Park, H. Y., Su, K. C., Katz, Z., English, B. P., Dey, A., Pan, K., Cheeseman, I. M., Singer, R. H., and Bathe, M. (2015). Inferring transient particle transport dynamics in live cells. *Nature methods*, *12*(9), 838–840.

Moretto, E., Miozzo, F., Longatti, A., Bonnet, C., Coussen, F., Jaudon, F., Cingolani, L. A., and Passafaro, M. (2023). The tetraspanin TSPAN5 regulates AMPAR exocytosis by interacting with the AP4 complex. *eLife*, *12*, e76425.

Opazo, P., Labrecque, S., Tigaret, C. M., Frouin, A., Wiseman, P. W., De Koninck, P., and Choquet, D. (2010). CaMKII triggers the diffusional trapping of surface AMPARs through phosphorylation of stargazin. *Neuron*, *67*(2), 239–252.

Opazo, P., Sainlos, M., and Choquet, D. (2012). Regulation of AMPA receptor surface diffusion by PSD-95 slots. *Current opinion in neurobiology*, *22*(3), 453–460.

Patterson, M. A., Szatmari, E. M., and Yasuda, R. (2010). AMPA receptors are exocytosed in stimulated spines and adjacent dendrites in a Ras-ERK-dependent manner during long-term potentiation. *Proceedings of the National Academy of Sciences of the United States of America*, *107*(36), 15951–15956.

Patterson, M., and Yasuda, R. (2011). Signaling pathways underlying structural plasticity of dendritic spines. *British journal of pharmacology*, *163*(8), 1626–1638.

Saxton M. J. (2007). A biological interpretation of transient anomalous subdiffusion. I. Qualitative model. *Biophysical journal*, *92*(4), 1178–1191.

Tan, J. Z. A., Jang, S. E., Batallas-Borja, A., Bhembre, N., Chandra, M., Zhang, L., Guo, H., Ringuet, M. T., Widagdo, J., Collins, B. M., and Anggono, V. (2023). Copine-6 is a Ca^2+^ sensor for activity-induced AMPA receptor exocytosis. *Cell reports*, *42*(12), 113460.

Tokunaga, M., Imamoto, N., and Sakata-Sogawa, K. (2008). Highly inclined thin illumination enables clear single-molecule imaging in cells. *Nature methods*, *5*(2), 159–161.